# Fertilization controls tiller numbers via transcriptional regulation of a *MAX1*-like gene in rice cultivation

Jinying Cui[1], Noriko Nishide[1], Kiyoshi Mashiguchi [2], Kana Kuroha[3], Masayuki Miya [1], Kazuhiko Sugimoto[3,4], Jun-Ichi Itoh[1], Shinjiro Yamaguchi [2] & Takeshi Izawa [1] ✉

Fertilization controls various aspects of cereal growth such as tiller number, leaf size, and panicle size. However, despite such benefits, global chemical fertilizer use must be reduced to achieve sustainable agriculture. Here, based on field transcriptome data from leaf samples collected during rice cultivation, we identify fertilizer responsive genes and focus on *Os1900*, a gene orthologous to *Arabidopsis thaliana MAX1*, which is involved in strigolactone biosynthesis. Elaborate genetic and biochemical analyses using CRISPR/Cas9 mutants reveal that *Os1900* together with another *MAX1*-like gene, *Os5100*, play a critical role in controlling the conversion of carlactone into carlactonoic acid during strigolactone biosynthesis and tillering in rice. Detailed analyses of a series of *Os1900* promoter deletion mutations suggest that fertilization controls tiller number in rice through transcriptional regulation of *Os1900*, and that a few promoter mutations alone can increase tiller numbers and grain yields even under minor-fertilizer conditions, whereas a single defective *os1900* mutation does not increase tillers under normal fertilizer condition. Such *Os1900* promoter mutations have potential uses in breeding programs for sustainable rice production.

Plant growth requires nutrition in the form of nitrogen (N), phosphorous (P), potassium (K), and minerals incorporated mainly from the soil, as well as carbohydrates obtained through photosynthetic activity[1]. Although the development of chemical fertilizers has led to increased crop yield, that of major cereal crops is limited by lodging caused by overgrowth. In breeding programs of the Green Revolution (1940–1960s), genetic improvement through the use of semi-dwarf traits successfully boosted lodging resistance in wheat and rice, achieving historic yield increases that saving many lives[2–5]. These semi-dwarf traits were obtained through a few genetic changes in the biosynthesis or signaling pathways of gibberellin (GA) phytohormones. However, the overuse of chemical fertilizers has led to aquatic and terrestrial pollution, including water eutrophication and soil consolidation[5]. The application of chemical fertilizer in agriculture also requires advance investment prior to income realization. Thus, chemical fertilizer-based agriculture is unsustainable, and more sustainable agricultural methods to reduce the demand of chemical fertilizer are sought globally[6].

Fertilization results in biological changes in many plant growth parameters, particularly increases in tiller (i.e., cereal branch) number. This is closely related to panicle number, leaf size, and panicle size, all of which help improve cereal yield[6]. Fertilizer regulates branching via mediating the phytohormone balance[1]. The phytohormone of strigolactone (SL), which can be induced in poor soil[7], is also involved in

[1]Lab. of Plant Breeding & Genetics, Department of Agricultural and Environmental Biology, The University of Tokyo, Tokyo, Japan. [2]Chemistry of Molecular Biocatalysts Lab, Institute for Chemical Research, Kyoto University, Gokasho, Uji, Kyoto, Japan. [3]Breeding Material Development Unit, Basic Research Division, National Institute of Crop Science, Tsukuba, Ibaraki, Japan. [4]Present address: Division of Crop Design Research, Institute of Crop Science, Tsukuba, Ibaraki, Japan. ✉e-mail: takeshizawa@g.ecc.u-tokyo.ac.jp

branching regulation[1]. Moreover, SL biosynthesis- and signaling-deficient mutants e.g., *d17*, *d27*, *d3* in rice displayed insensitivity to nutrient depletion[8], while mutants for either the SL−biosynthesis or SL−signaling pathway contributed significantly to the increase in branching or tiller numbers[9–11].

Inorganic P is often presented as an insoluble complex in soil, which greatly hinders the accessibility of P to plants[12,13]. Therefore, apart from the direct uptake of soluble P by roots[14], plants such as rice and soybean have developed an indirect absorption method, i.e., P transfers from mycorrhizal hyphae into root cortical cells to ensure sufficient supply[15,16]. For such an indirect nutrient uptake, SLs act as signaling molecules to establish necessary symbiosis between the host plant and arbuscular mycorrhizal (AM) fungi[17]. As reported, many genes synergistically promote SL biosynthesis in rice via a multi-step biochemical reactions including the formation of carlactone (CL) from all-*trans*-β-carotene and its further transformations to carlactonoic acid (CLA), 4-deoxyorobanchol (4DO), and orobanchol (ORO, a hydroxylated 4DO)[18,19].

A rice genomic region, which contains deletions of both *Os900* (*Os01g0700900*) and *Os1400* (*Os01g0701400*) genes that are homologous to *MORE AXILLARY GROWTH1* (*MAX1*) gene in *A. thaliana*, was identified as a major quantitative trait locus (QTL) influencing tiller number and SL-derivative biosynthesis[20]. *MAX1* gene encodes a cytochrome P450 enzyme for SL biosynthesis to regulate branching in *A.thaliana*[9]. It has been known that *MAX1* gene and its orthologs play essential roles in the transformation of CL in many plants[21]. Unlike dicotyledons, however, such as *A. thaliana* and tomato possessing only one *MAX1* gene for the conversion of CL into CLA in their genomes[9,22,23], multiple *MAX1* genes exist in monocotyledons like rice and maize[18]. The rice genome consists of five *MAX1*-like genes, *Os900*, *Os1400*, *Os1500* (*Os01g0701500*), *Os1900* (*Os02g0221900*), and *Os5100* (*Os06g0565100*), while *Os1500* among them is considered as a loss-of-function allele according to the DNA sequence data[18,19]. Until very recently, no study besides the above QTL analysis has been carried out to analyze mutations of these *MAX1*-like genes in rice. Based on an extensive analysis of *os900* mutants, Ito et al. (2022) strongly suggest that canonical SLs such as 4DO are important rhizospheric signals but not the major determinant of tillering[24]. Thus, how rice each *MAX1*-like gene is involved in rice tillering remains an open question in this field. It is also reported that the P starvation can induce the expression of *Os900* and *Os1900* (but not for *Os1400* and *Os5100*)[25], which *Os900* and *Os1400*, as the QTL genes, facilitate the biochemical CL-to-CLA conversion and the further biosynthesis from CLA to 4DO (by *Os900*) and from 4DO to ORO (by *Os1400*) in vitro[18,19]. By contrast, *Os1900* and *Os5100* transfer only small amount of CL into 4DO in vitro[19]. Apparently, *Os900* and *Os1400* rather than *Os1900* or *Os5100* are considered as major genetic contributors so far in the biosynthesis of 4DO and ORO in rice.

In this study, we show that the fertilizer-responsive *MAX1*-like gene, *Os1900*, cooperating with *Os5100*, contribute to the CL-to-CLA conversion to regulate tillering in rice. Furthermore, the study on a series of *cis*-regulatory variations in *Os1900* reveals that fertilizer mediates tillering through the transcriptional regulation of *Os1900* and such artificial promoter variations have great potential for the agricultural applications, which will lay the foundation for breeding novel rice varieties with low fertilizer demand and high production yield.

## Results and discussion

To elucidate the molecular mechanisms of fertilizer response in rice in real paddy field cultivation, we first performed field transcriptome analysis of leaf samples from rice plants that were grown under three fertilization treatments (None, Once, Twice, as shown in Fig. 1a) in adjacent paddy fields, and analyzed the frequency distribution of panicle and seed numbers of tested plants. Rice panicle numbers and

grain yields increased clearly in association with additional fertilization treatment (Fig. 1b and Supplementary Fig. 1a, b). To confirm the effects of fertilization on gene expression in response to environmental factors in rice cultivation, we collected leaf samples on eight different dates after transplanting for transcriptome analysis (at 10:00 and 16:00 randomly during the sampling terms, see details in Fig. 1a, Supplementary Data 1, 2). Accordingly, these transcriptome data showed broad variations due to fluctuating environmental factors such as ambient temperature and solar radiation, as well as endogenous factors e.g., growth stage and circadian clock phase. A principal component analysis (PCA), which was conducted using these transcriptome data, indicated that the factors including sampling time and transplantation time besides fertilization had the greatest influence on the transcriptome (Supplementary Fig. 1c). Thus, these data encompass the effects of all environmental factors as well as fertilization. Accordingly, in this transcriptome data the effects of fertilizers were relatively low against the fluctuations of gene expression due to various environmental stimuli in nature, which can be represented by the standard deviations of those data. Then, in order to only highlight effects of fertilizer using such data from the real rice cultivations, we plotted the expression fold changes against fertilization (FCs) versus their ratios (FC/SDs) to the standard deviation (SD) of gene expression among all tested samples for each gene (Fig. 1c). Genes with high FCs and high FC/SD ratios can be further identified as candidate genes related to fertilizer effects (Fig. 1c). With these fluctuating data, we repeated simple paired *t*-test using only different fertilization conditions for many sample combinations (Supplementary Data 1, 3). These paired *t*-tests suggest that 107 genes are stable fertilizer-responsive genes in rice leaves under natural paddy field conditions (Fig. 1d, Supplementary Data 3).

Considering the annotation information for the identified 107 genes and the FC/SD ratio, we focused on *Os1900*, a rice *MAX1*-like gene involved in SL biosynthesis (Fig. 1c, d) since we came up with an idea in which fertilizer-controlled transcript of *Os1900* can be a key to explain the relationship between fertilizer and tillering in rice. It is known that some of SL-related genes, e.g., *D10*, *D17*, *D27*, *Os900* and *Os1900*[7], respond to changes in the concentration of the major plant nutrient of P[25]. However, only *Os1900* expression is repressed by fertilization in the leaf samples under paddy field cultivation conditions, where their tissue specific expression might be involved. A question then arises: does fertilization control tiller numbers via *Os1900* in rice? In addition to *Os1900*, other *MAX1*-like genes such as *Os900*, *Os1400*, and *Os5100* were reported to restore *max1* mutant in *A. thaliana* to the wild type phenotypes when artificially expressed[20,26]. Since *Os5100* is more closely homologous to *Os1900* than the other *MAX1*-like genes[18], CRISPR-Cas9 mutants of *Os1900* and *Os5100* including *os1900* single, *os5100* single and *os1900&os5100* double mutant (Fig. 2a; Supplementary Fig. 2a–c) were generated. Unexpectedly, we only observed significant increase of tillers in *os1900&os5100* double mutant plants under normal fertilization conditions, rather than in the *os1900* or *os5100* single mutant plants (Fig. 2c, d). Notably, several independent $T_0$ mutants having *os1900&os5100* double mutations exhibited similar phenotypes (Supplementary Fig. 2d). To exclude any unexpected mutations through off-target CRISPR/Cas9 action, we examined k-mer coverages using rice genome DNA sequences (IRGSP1.0) as references for all *MAX1*-like gene regions of the *os1900&os5100* double mutations. The k-mer gaps indicated that mutations occurred only in *Os1900* and *Os5100* instead of in *Os900*, *Os1400*, or *Os1500* (Fig. 2b).

Given that environmental factors such as ambient temperature and radiation often influence axillary bud dormancy (or elongation), branching, or tiller (branch) numbers under natural cultivation conditions[27], we cultivated the wild-type (WT, cv. Koshihikari) and *os1900&os5100* double mutant plants under four different photoperiod and ambient temperature conditions, which showed consistent phenotypes for each condition (Fig. 2e) in spite of differences in tiller

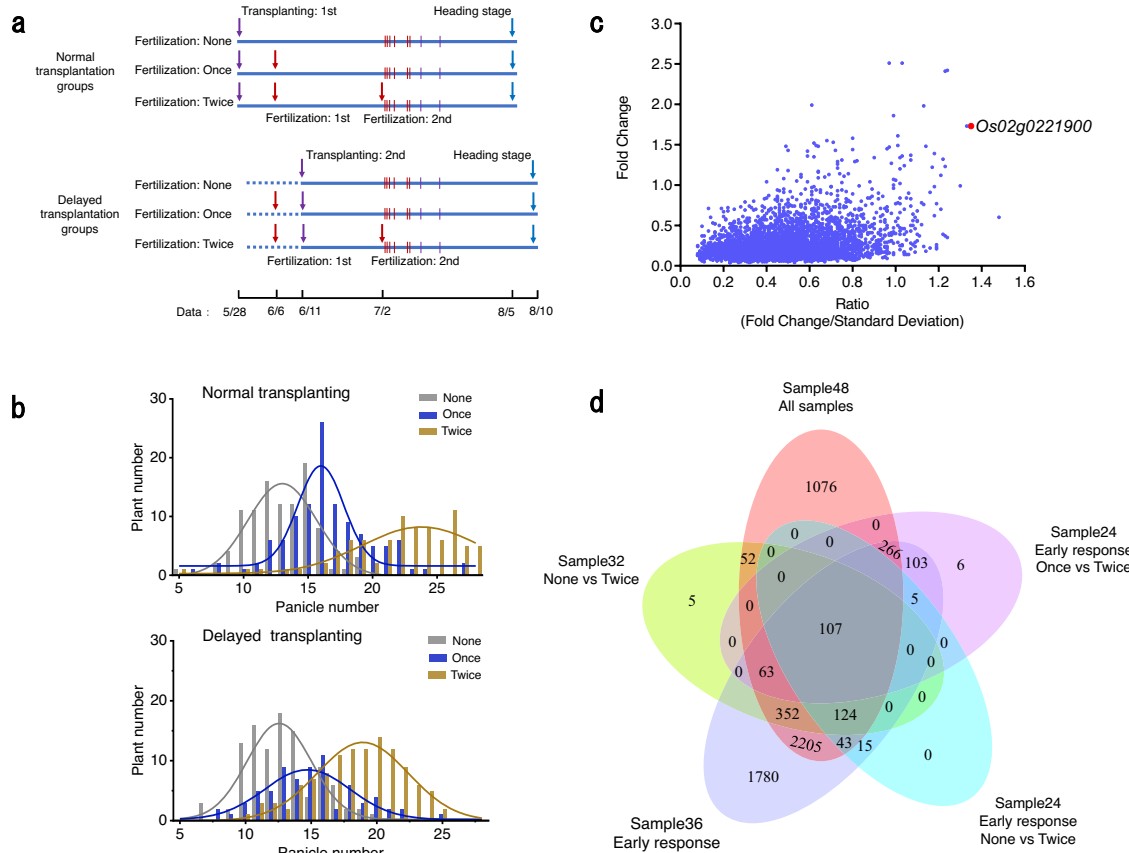

**Fig. 1 | Rice gene expression in response to fertilization under natural paddy field conditions. a** Schedule of fertilization and sampling time for transcriptome analysis. Vertical line: Sampling timing, 16:00 7/2, 10:00 7/3, 16:00 7/3, 10:00 7/5, 10:00 7/8, 16:00 7/8, 16:00 7/11, 16:00 7/16. Red ones: early response sampling. Purple ones: late response sampling. See Supplementary Data 1. **b** Distribution of plants according to panicle numbers per plant under normal and delayed transplanting conditions in the paddy field, Tsukuba, Japan. **c** Gene expression in response to fertilization under fluctuating environmental conditions according to field transcriptome analysis. Fold change = abs (Mean of $\log_2$ (gene expression without fertilization/gene expression with fertilization in a paired sample for each gene)), Ratio = Fold Change/Standard Deviation. This ratio indicates the contribution of fertilization against the entire fluctuation by surrounding environmental factors such as radiation and ambient temperature during cultivation for each gene. **d** Venn diagram of gene expression under different fertilization combinations such as early responses and late responses according to field transcriptome analysis; FDR (false discovery rate) < 0.01.

numbers among treatments. Thus, these tiller number phenotypes were very stable in the *os1900&os5100* double mutants. In addition, the double mutants produced more secondary and tertiary tillers instead of primary tillers than WT plants. After one month of sowing without fertilization, however, even the *os1900&os5100* double mutants exhibited no increase in tiller numbers (Supplementary Fig. 3a, b). Since it is known that the lack of nitrate reduces shoot branching in *A. thaliana*[28], in the absence of nutrient uptake, the rice plants could not execute axial bud elongation anymore.

As described above, *Os900* and *Os1400* were previously identified as major QTLs using Azucena and Bala as parent cultivars[20], which strongly implies that they are the major genetic contributors to tillering in rice. To solve this controversy, we further generated the *os900&os1400* double mutants and compared its phenotypes with *os1900&os5100* double mutants (Supplementary Fig. 4a) under normal fertilizer conditions. We found a considerably sharper increase of tiller numbers in *os1900&os5100* double mutants than that in *os900&os1400* mutants (Supplementary Fig. 4b, c). Even under minor and trace fertilization conditions, *os900&os1400* displayed almost same tiller number with wild type (Supplementary Fig. 4d). This indicates that tiller number was mainly regulated by *Os1900* & *Os5100*, although *Os900* and *Os1400* together contributed to the control of maintained natural variations in tiller number among the rice cultivars.

Recent next-generation sequencing analysis of various rice genomes has revealed that the region carrying *Os900* and *Os1400*

represent more likely an insertion in an ancient *Oryza sativa* subsp. *japonica* accession, rather than a deletion in another ancient *Oryza sativa* subsp. *indica* accession[29]. Because the induced expression of *Os900* by P starvation in roots resembles that of *Os1900* in shoots[25], this insertion could be an adaptation for enhanced P acquisition[29,30] by inducing hyphal branching of arbuscular mycorrhizal fungi[17]. Taking these into consideration, *Os1900* and *Os5100* may have been originally involved in the regulation of tiller dormancy and subsequent elongation with response to fertilization.

The previous transient expression assay using *Nicotiana benthamiana* revealed more enzymatic activity of Os900 and Os1400 proteins than those of Os1900 and Os5100 proteins in the biosynthesis of 4DO and ORO (Fig. 3a)[19]. Os1900 and Os5100 proteins could convert only a small amount of CL into 4DO[19]. In addition, the CL feeding assay using recombinant proteins in yeast suggested that Os1900 and Os5100 can catalyze the conversion of CL to CLA[18]. However, the genetic roles of *Os1900* and *Os5100* in SL biosynthesis remain elusive in rice.

In this context, we performed liquid chromatography–tandem mass spectrometry (LC–MS/MS) to measure the SLs content in the extracts from shoots and roots of WT (a cultivar termed Koshihikari) and *os1900&os5100* double mutant from young rice seedlings (Fig. 3b, c and Supplementary Fig. 5a, b).

Tremendous CL accumulation was observed in both shoots and roots in *os1900&os5100* double mutants (Fig. 3b, c). The accumulation

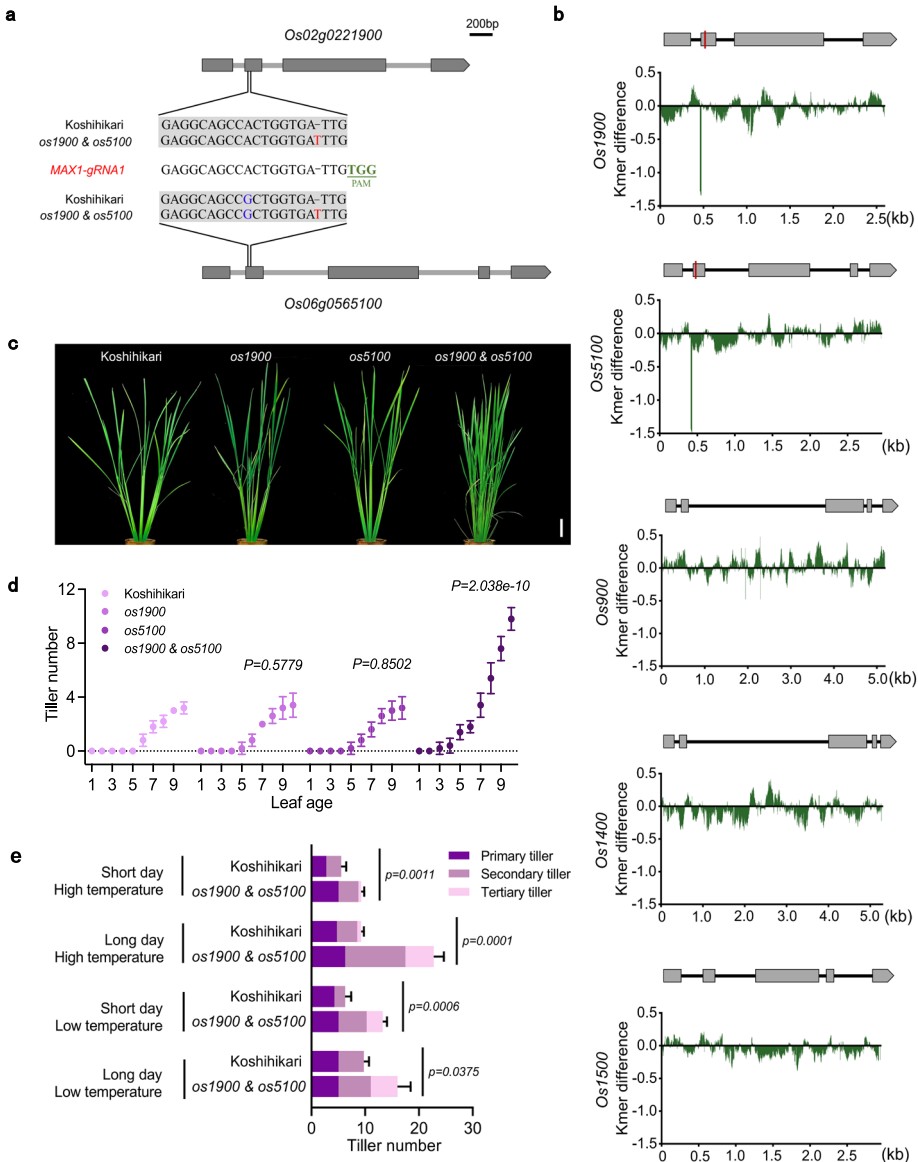

**Fig. 2 | Phenotypic analysis and genotype identification results for the os1900&os5100 mutant. a** *MAX1*-gRNA1 and its target sequence and mutation sites. The single *MAX1*-gRNA1 caused mutations in both *Os1900* and *Os5100*. **b** k-mer coverage differences among the *Os1900*, *Os5100*, *Os900*, *Os1400*, and *Os1500* genes in wild-type (WT, cv. Koshihikari) and os1900&os5100 mutant in rice. Vertical red lines indicate the locations of gRNA used in this experiment. y-axis: log₁₀ (k-mer for os1900&os5100/k-mer for WT). The k value was 20. **c** Photographs of tiller phenotypes including WT and *os1900* and *os1500*, and *os1900&os5100*

mutants. Bar = 5 cm. **d** Tiller numbers from the 1-leaf to 10-leaf stages. The P-values for different pairs (Koshihikari vs. *os1900*; Koshihikari vs. *os5100*; Koshihikari vs. *os1900&os5100*) were obtained based on making generalized linear mixed models (GLMMs) and ANOVA analysis. Error bars indicate SD, $n = 5$ biologically independent plants. **e** Comparison of tiller number between WT and the os1900&os5100 mutant under four cultivation conditions, $n = 4$ biologically independent plants. Significance values are from Student's $t$-test (two-tailed). Error bars: SD.

in shoots was up to approximately 92,000 pg/g fresh weight, which was significantly superior to that in WT (Fig. 3b), while only tiny amount of CLA was detected in os1900&os5100 double mutants. Notably, the detected amount of CL in shoots (Fig. 3b) exceeds that in roots (Fig. 3c) in the os1900&os5100 double mutants with almost triple digit difference. *MAX1* gene is a crucial gene to convert CL into CLA in *A.thaliana*, the same as the reported activity of *Os900* and *Os1400* in rice[18,19,22]. When *MAX1*-like transcripts were examined, *Os900* and *Os1400* genes exhibited the same transcriptional levels in os1900&os5100 mutants as those in WT in both shoots and roots (Fig. 3d, e), while *Os1900* and *Os5100* genes exhibited significantly lesser transcriptional levels in os1900&os5100, may be due to the incomplete nonsense-mediated mRNA decay (Fig. 3d, e). Thus,

different accumulation of CLs between shoots and roots can be ascribed to the higher expression of *Os900* and *Os1400* in the roots, thus resulting in higher consumption of CL in roots[25] (Fig. 3d, e). Accordingly, the loss of function mutations of *Os1900* and *Os5100* directly causes the accumulation of CL in shoots and roots of os1900&os5100 mutants.

Small amount of 4DO was detected in shoots of the os1900&os5100 double mutants while no 4DO was detected in WT shoots (Supplementary Fig. 5a). Although there is no difference of *Os900* (which converts CL to 4DO[18]) expression at shoot samples (Fig. 3d), it was induced at stem base and tiller base in os1900&os5100 mutants (Supplementary Fig. 5c, d). This might result in the small amounts of 4DO in shoots of the double mutants. Furthermore, larger

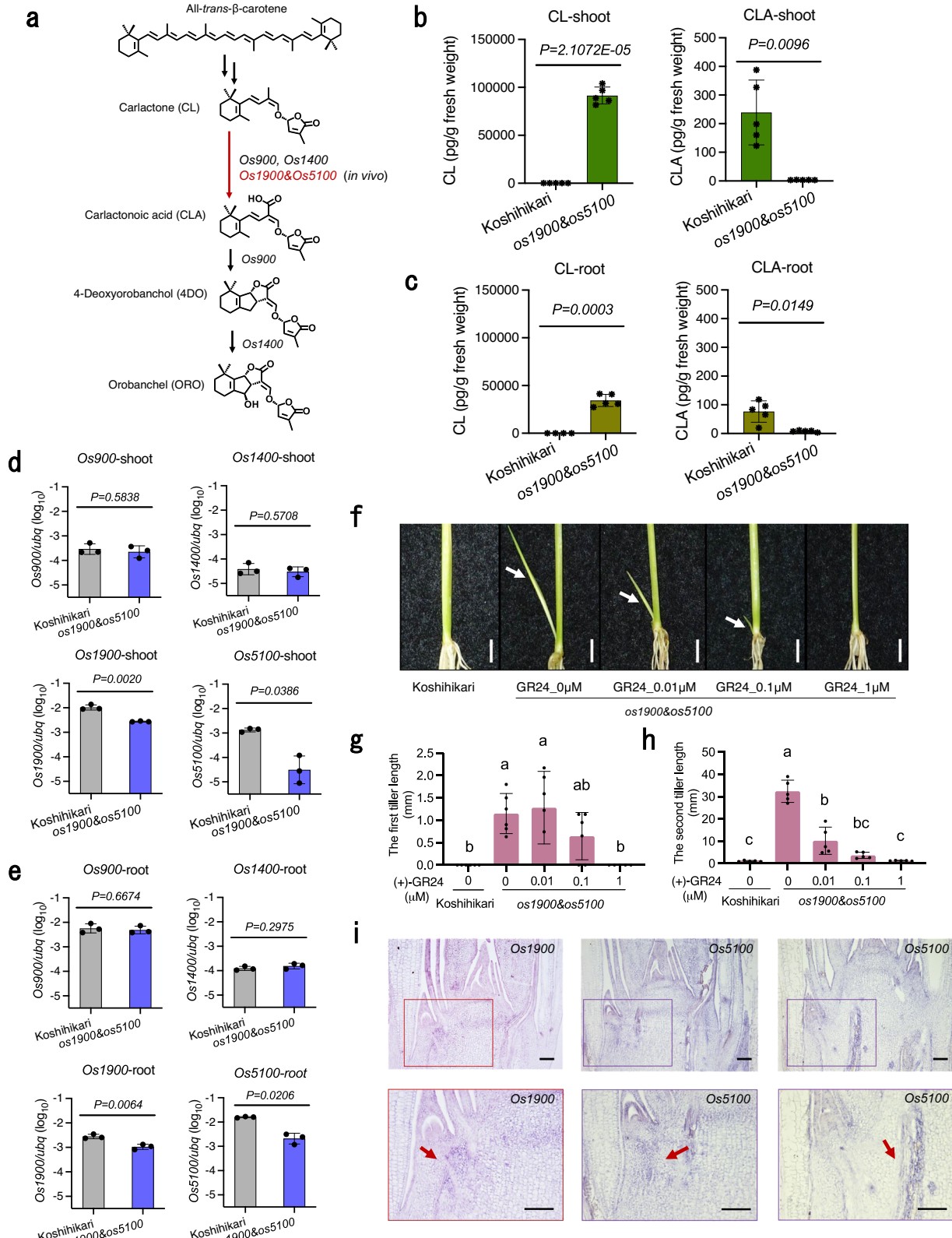

amounts of 4DO were detected in the roots of the *os1900&os5100* double mutants than that of WT (Supplementary Fig. 5b), despite no induced expression of *Os900* (Fig. 3e). This agrees with the phenomenon that Os900 protein facilitates CLA carboxylation into 4DO via intramolecular cyclization in vitro[18,19], as *Os900* expression was detected predominantly in roots[25]. It is possible that Os900 can immediately produce 4DO from accumulated CL, and the

intermediate, CLA, is not sufficiently released during this conversion. Taken together, the higher level of 4DO in the roots of *os1900&os5100* double mutants may be due to the conversion of the excessively accumulated CL to 4DO by Os900 in the roots. No ORO, which is a hydroxylated 4DO catalyzed by Os1400 as the final product of proposed SL biosynthetic pathway[18,19] (Fig. 3a), was detected in the WT or *os1900&os5100* double mutants in this study. Pairwise sequence

**Fig. 3 | Analyses of strigolactone (SL) biosynthesis in WT and *os1900&os5100* mutant plants. a** Proposed SL biosynthesis pathway in rice. Genes name marked in black are previous works[18,19], that in red is this work. **b, c** Endogenous levels of carlactone (CL) and carlactone acid (CLA) in shoot extracts (b) and root extracts **c**, *n* = 5 biologically independent samples. **d, e** Gene expression analysis of *Os900*, *Os1400*, *Os1900* and *Os5100* in shoot **d** and root **e** of Koshihikari and *os1900&os5100* mutants sampled in the same manner as plant used for LC-MS/MS, n = 3 biologically independent samples. Error bar is SD, Significance values are from Student's *t*-test (two-tailed) **b–e**. **f–h** Tiller length in WT and *os1900&os5100* mutants under different concentrations of a synthetic SL analog ((+)-GR24), unit: μM. White arrows indicate the second tiller. Bar in **f** represents 0.5 cm. In **g**, **h**, the error bar indicates SD, *n* = 6 biologically independent plants, multiple comparisons, turkey (*P* < 0.05), *P*-value see Supplementary Data 4. **i** Gene expression of *Os1900* and *Os5100* in plants produced by in situ hybridization. Bar = 100 μm. Two times the experiment was repeated with similar results. For histone *H4* as a positive control of this experiment, see Supplementary Fig. 5e.

alignment in *Os1400* between Azucena, an ORO-producing cultivar[20] and Koshihikari showed a single-nucleotide polymorphism (SNP), G-to-A in the exon 5, that results in the R-to-Q substitution of the corresponding amino acid (Supplementary Fig. 6a). This R in this corresponding amino acid site is highly conserved in *MAX1* homologous genes among the plant kingdom (Supplementary Fig. 6b, c); and we only found the SNP change from R to Q among *Oryza sativa* subsp. *japonica* cultivars including Koshihikari (Supplementary Fig. 6d), suggesting that the R-to-Q amino acid substitution may result in a weak enzymatic activity of the Koshihikari-type of Os1400 protein.

Methyl carlactonoate (MeCLA) that is produced through CLA methylation in *A. thaliana*[22], was under the detection limit in the shoots and the roots of *os1900 & os5100* (Supplementary Fig. 5a, b), which may be because of the absence of its precursor, CLA.

As reported in other known SL biosynthesis mutants[31], axillary bud elongation in *os1900&os5100* was inhibited by the application of the synthetic SL analog (+)-GR24, and the inhibitory effect was weak under low concentration (Fig. 3f–h), which reminds that although small amounts of 4DO were detected in shoots of the *os1900&os5100* double mutants (Supplementary Fig. 5a), the *os1900&os5100* double mutants produced more tillers than WT, indicating that those small amounts of 4DO could not rescue the higher tiller number phenotype in the *os1900&os5100* double mutants compared to WT.

To reveal the genome-wide effects of the key *MAX1*-like genes to establish the tillering phenotypes in this study, i.e., genetic roles of *Os1900* and *Os5100* in transcriptome and their potential physiological roles, RNA-seq analysis was performed on leaves and tiller bases of WT and *os1900&os5100* (Supplementary Fig. 7-9). Plenty of differentially expressed genes (DEGs) in both tiller bases and leaves were identified by QLF test in edge R with FDR < 0.01 (Supplementary Fig. 7b and Supplementary Data5). The mean of log2FC values between WT and the *os1900&os5100* double mutants were used in graphs (Supplementary Fig. 7-9). Some key genes expression was confirmed with real-time qPCR using same samples (Supplementary Fig. 7, 10). With *os1900 & os5100* double mutations, which have blocked conversion from CL to CLA, inhibiting the production of active SLs, we first revealed that the genetic loss in *Os1900 & Os5100* decreased the expression of *D53*, a key repressor gene of SLs signaling, in both tiller bases and leaves (Supplementary Fig. 7c). However, the expression of *D14*, a SL receptor that 4DO can bind to, displayed slightly higher level in *os1900&os5100* double mutants (Supplementary Fig. 7d), which can be assigned to a feedback enhancement of SL sensitivity (Supplementary Fig. 7c) and *Os900*-induced production of more 4DO (consistently with data in Supplementary Fig. 5). Besides, the rice *Teosinte Branched 1* (*OsTB1*), a negative regulator of branching, was down-regulated with only half fold difference in *os1900&os5100* double mutants (Supplementary Fig. 7d), which is consistent with the insensitivity of *tb1* mutants to GR24[32]. In this context, the drastic CL accumulation accompanied with the production of trace SLs in *os1900&os5100*, have associations (or influences) on the expression of genes related to SL biosynthesis and signal transduction.

Unexpectedly, we further found transcriptional regulations related to biosynthesis and signaling of other phytohormones by *Os1900 &Os5100* based on the RNA-seq data. Both cytokinins (CKs) and SLs are second messengers of auxin that can regulate plant architectures. SLs interfere with the formation of auxin-conducting channels by inhibiting the intracellular accumulation of PINs in *A.thaliana*[33]. In our data, the expression of these auxin transporter genes, i.e., *PIN FORMED5a* (*PIN5a*), *PIN9*, and *AUXIN4* (*AUX4*) were down-regulated in the tiller bases of *os1900&os5100* (Supplementary Fig. 8a). We found induced expression of CKs-biosynthesis related genes (i.e., *LONELY GUY 1*(*LOG*) and *LOG-Like* genes) and a CKs-signaling related gene (i.e., *A-TYPE RESPONSE REGULATOR 2* (*RR2*)) in tiller bases or leaves in *os1900 & os5100* double mutant, while the expression of CKs-inactivation- related genes (i.e., *cytokinin oxidases* (*CKXs*)) was repressed (Supplementary Fig. 8b), which was consistent with that CKs is the hormone for promoting tiller elongation[34].

The key genes related to ABA and JA signal transduction (i.e., *Serine/threonine protein kinases* (*SAPKs*), *a type 2 C protein phosphatase* (*PP2C*), and *JASMONATE-ZIM*-domain (*JAZ*) proteins) were up-regulated in *os1900&os5100* (Supplementary Fig. 8c,d), indicative of controlled axillary bud dormancy by SLs via mediating the ABA and JA signaling pathways in rice[35]. Furthermore, the expression of genes related to biosynthesis and signaling of salicylic acids (SAs) was induced thus promoting the $CO_2$ fixation[36] (Supplementary Figs. 8g and 9a). Taken together, differentially expressed genes related to CKs, auxins biosynthesis and signaling pathways; ABA signal pathway; JA biosynthesis pathway behaved in the same manner in tiller bases as that in leaves (Supplementary Fig. 7, 8). These results clearly revealed the central role of *Os1900 & Os5100* in controlling most of phytohormone signaling networks to control tiller dormancy and elongation in rice.

In the previous studies, it has been shown that SLs are synthesized in roots and transported upward to the stem bases to inhibit branching (or tillering)[9,37]. In this study, *Os1900* and *Os5100* clearly are expressed in shoot, e.g., stem base and tiller base (Fig. 3d, Supplementary Fig. 5c, d). The in situ expression analysis of *Os1900* and *Os5100* reveals their transcripts around axial bud apex regions and a slightly distinct expression pattern of *Os5100* from those of *Os1900* near vascular tissues around apex regions (Fig. 3i). Together, these findings imply that the expression pattern of in vivo Os1900 and Os5100 proteins at tiller bases, corresponding to the distribution of genuine MAX1 enzymatic activity and the spatial transformation of CL to CLA including possible transportation of SL derivatives from shoots needs to be further revealed to understand detailed regulation of tillers in rice[22,23,38].

*FT-like* genes i.e., *Hd3a* in rice[39] and *PtFT1* in *Populus tomentosa*[40,41] could break axillary bud dormancy and promote the lateral branching. In our data, a *FT-like* gene of *FTL6* had a higher expression level in leaves in *os1900&os5100* than that in WT (Supplementary Fig. 9b). Thus, there was a possibility that *FTL6* was also involved in breaking axillary bud dormancy in rice. In addition, since *OsMFT1* and *OsMFT2*, could break seed dormancy[42], it is also possible that the increased expression of *OsMFT1* in both leaves and tiller bases in *os1900&os5100* are involved in the tiller number phenotypes (Supplementary Fig. 9b).

Although *Os1900* was responsive to fertilization, the tiller numbers did not increase for *os1900* single mutant under normal experimental conditions (Fig. 2c, d). Thus, we conducted a detailed gene expression analysis of *Os1900* expression under laboratory conditions (Fig. 4a). More specifically, fertilizer was applied once on the 7th day after sowing. *Os1900* expression remained at the baseline level until the 13th day after sowing, when fertilizer concentration (in terms of total dissolved solids, TDS) gradually decreased down to ~60 ppm

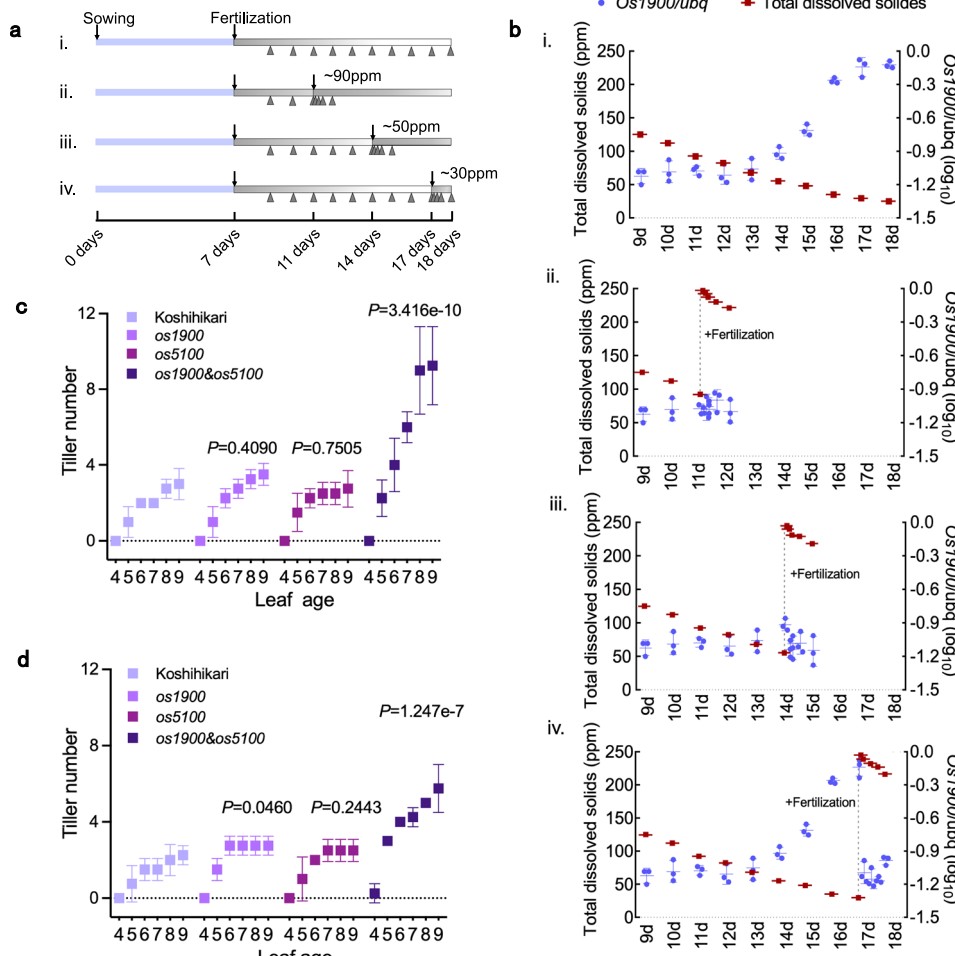

**Fig. 4 | Detailed gene expression of *Os1900* to fertilization and regulation of tiller number under minor- and trace- fertilization conditions. a** Schematic diagram of fertilization conditions for *Os1900* expression analysis in leaves. After germination, plants were cultivated without fertilizer for 7 days, and then ½ Kimura's B solution fertilizer was applied. From the 9th day, group i was sampled daily, and groups ii-iv were fertilized again when fertilizer concentration (total dissolved solids, TDS) dropped to 90, 50, and 30 ppm, respectively. Sampling (triangles) was conducted at 0, 1, 3, 6, and 24 h after fertilization. **b** Analysis of *Os1900* expression under different fertilizer conditions. The error bar is SD, *n* = 3, 4 biologically independent samples. **c, d** Tiller numbers of *os1900*, *os5100* single mutant and *os1900&os5100* double mutant under minor- **c** and trace- **d** fertilization condition from 4- to 9-leaf age, *n* = 5 biologically independent plants. Cultivation condition: see methods. The *P*-values for different pairs (Koshihikari vs. *os1900*; Koshihikari vs. *os5100*; Koshihikari vs. *os1900&os5100*) are obtained by making a GLM model and subsequent ANOVA test. Error bars: SD.

(Fig. 4b-i). From the 14th day, *Os1900* was gradually expressed as TDS concentration decreased (Fig. 4b-i). Thus, *Os1900* expression was strongly repressed under moderate fertilization conditions in young seedlings, which gradually expressed in response to a decrease in soluble fertilizer concentration. When TDS concentration was reduced to 30 ppm, *Os1900* expression raised significantly compared to that at higher TDS concentration (Fig. 4b-i). For the individuals with high expression of *Os1900*, fertilizer treatment resulted in rapid decrease in *Os1900* expression to basal levels within 1 h (Fig. 4b-ii, -iii, -iv). The expression dynamics of *Os1900* under the laboratory conditions implied that its biological role could be clarified at lower-than-normal fertilization conditions. Therefore, we examined the tiller number phenotypes of *os1900* single, *os5100* single, and *os1900&os5100* double mutants by using minor- and trace-fertilization conditions. We only found a mild increase in tiller numbers for the single *os1900* mutation upon fertilization in trace level (Fig. 4c, d), indicating that though *os1900* mutation resulted in no changes in the gene expression of *Os5100* (Supplementary Fig. 3c). Fertilizer responses of *Os1900* transcripts and the tillering phenotype in related mutants implies the importance of transcriptional regulation of *Os1900* to control tiller

number in rice. Thus, the identification of key cis-regulatory sites in the *Os1900* promoter related to complex fertilization responses would help to elucidate the molecular mechanisms of transcriptional regulation in fertilization responses and tiller number control in rice. Although CRISPR/Cas9 mutants generated by targeting protein coding regions often result in clear loss-of-function alleles and extreme biological phenotypes, promoter region mutations can confer a variety of related phenotypes[43]. In the cis-regulatory editing system, alleles with overlapped deletions show similar phenotypic effects[43] and hidden pleiotropic roles[44] in genes. Therefore, we next generated a string of deletion alleles covering the entire *Os1900* promoter region to identify the regulatory regions, which are responsible for tiller number control by fertilization. We designed 19 guide RNAs (gRNAs) to create this set of mutants; every three adjacent gRNA expression units were cloned into one vector construct, for a total of nine vectors (Fig. 5a, b), as each deletion mutant had a deletion of a few hundred bps in the promoter region (till 5,000 bp upstream regions from the translation start site) of the *Os1900* gene. In the first generation (T₀) of regenerated plants, plants with large deletions (> -100 bp) were selected by polymerase chain reaction (PCR). DNA sequence analysis of plants of the T₁

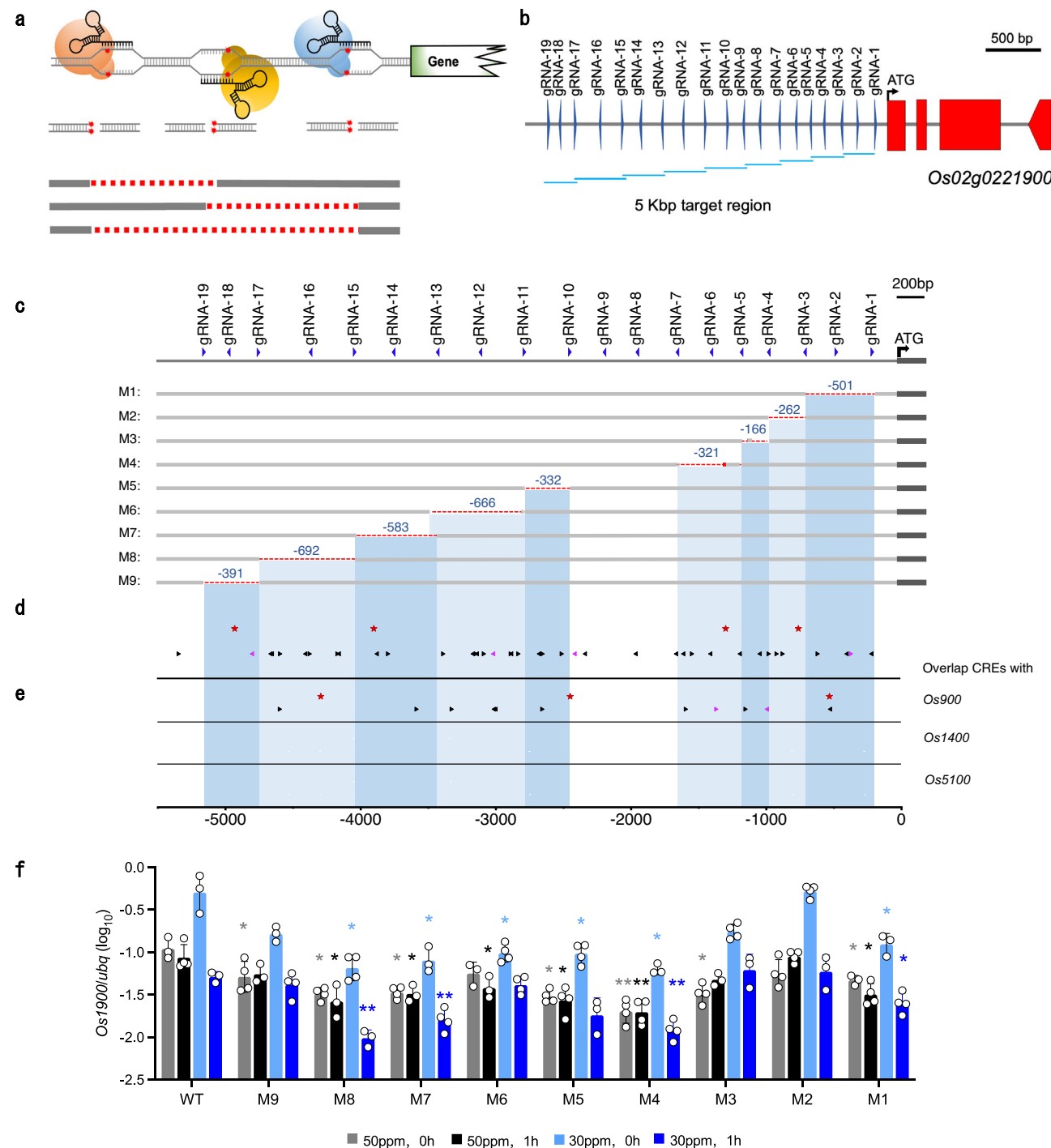

**Fig. 5 | Engineering of the *Os1900* promoter provided a continuum of alleles with different fertilization responses. a** Profile of three double-strand break sites and three expected deletion mutations. **b** Schematic view of the *Os1900* promoter target for 19 guide RNAs (gRNAs), where three gRNAs represent one construction. The target promoter region was 5 kb upstream of the translation start position (ATG). **c** Sequencing of *Os1900* promoter deletion mutants (*M1–M9*) among T₁ plants. (−) indicates deletions; (+) indicates insertions. **d** Potential fertilizer-related cis-regulatory elements (CREs) related to combinations of N, P-fertilizer in the *Os1900* promoter were scanned using the PlantPAN3.0 program at a relative profile score threshold of 99%. Black triangles: CREs related to P; Pink triangles: CREs related to N; Red star marks: P1BSs. **e** Overlapping evolutionarily conserved CREs of *Os1900* (showed in **d**) with other *MAX1*-like genes (*Os900*, *Os1400*, *Os5100*). **f** *Os1900* gene expression of various *Os1900* promoter mutants to different fertilizer concentrations, n = 3,4 biologically independent samples. Means of *M1–M9* were compared with the corresponding gene expression in WT in four conditions, respectively. 50 ppm 0 h and 1 h: TDS is 50 ppm and 245 ppm, respectively; 30 ppm 0 h and 1 h: TDS is 30 ppm and 245 ppm, respectively. Cultivation conditions and sampling time were the same with Fig. 5b. (*FDR < 0.05; **FDR < 0.01. FDR values, see Supplementary Data 4). Error bars indicate SD, Significance value is from Student's *t*-test (two-tailed).

generation identified nine alleles with distinct deletions, covering most of the entire target promoter region (Fig. 5c). Using these deletion mutants, we examined the fertilizer response of *Os1900* expression under four sampling conditions (i.e., 50 ppm 0 h and 1 h: TDS is 50 ppm

and 245 ppm, respectively; 30 ppm 0 h and 1 h: TDS is 30 ppm and 245 ppm, respectively).

To predict the evolutionarily conversed cis-regulatory elements (CREs), possible CREs related to N, P fertilizer (Fig. 5d, Supplementary

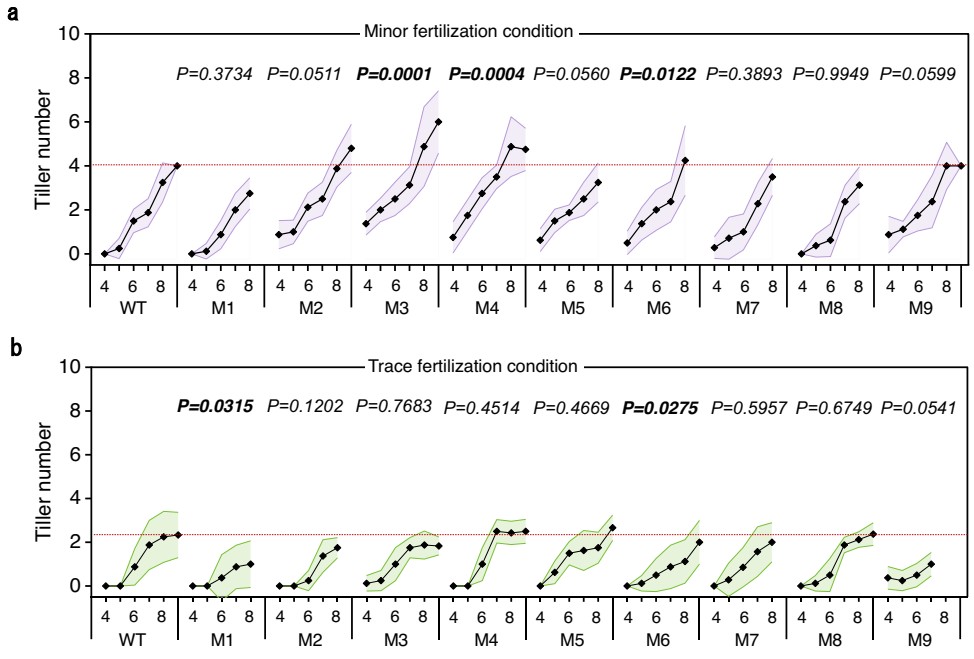

**Fig. 6 | Tiller numbers in a series of *Os1900* promoter mutant under minor-(a) and trace-(b) fertilization conditions.** Horizontal axis: leaf age. Fertilization condition: see methods. Data showed average values, $n = 8$ biologically independent plants. Purple **a** and green **b** shadows were error bands. The *P*-values for different pairs (Koshihikari vs. promoter deletion mutants) are obtained by making a GLMM and subsequent ANOVA test.

Data. 6) and the conserved motifs in promoter regions of other *MAX1*-like genes including *Os900*, *Os1400* and *Os5100* (Fig. 5e, Supplementary Data. 7) were surveyed. We found dozen of CRE candidate sites as evolutionarily conserved ones (Fig. 5d), meanwhile, CRE filtering was done in comparison with other *MAX1*-like genes (Fig. 5e). The major CREs for transcription factors (TFs) included members of the WRKY and MYB families, which include the main TFs of the AM symbiosis network[16]. *Os900* and *Os1900* were induced by manure starvation, while the *Os1400* and *Os5100* were insensitive to it. Accordingly, it might be reasonable that there are no fertilizer-related CREs predicted in promoter regions of *Os1400* and *Os5100*.

Then, *Os1900* expression levels for the four fertilizer conditions ((1) 50 ppm,0 h: TDS is -50 ppm; (2) 50ppm,1 h: TDS is -245ppm; (3) 30 ppm,0 h: TDS is -30 ppm; (4) 30 ppm,1 h: TDS is -245 ppm) are compared to those of WT in Fig. 5f. *Os1900* expression was unchanged by fertilization in WT (50 ppm, 0 h vs. 50 ppm, 1 h), even when the fertilizer concentration changed rapidly (Figs. 4b and 5f and Supplementary Data. 4). The intercept values (which indicate *Os1900* expression at 0 h) of all tested mutants were significantly lower than that of WT at 50 ppm, except *M2* and *M6* (Fig. 5f; Supplementary Fig. 11 and Supplementary Data 4 and 8). All mutants exhibited at most a very weak fertilizer response at 50 ppm; however, positive trends were observed in *M2* and *M3* (Supplementary Fig. 11; Supplementary Data 8). By contrast, at 30 ppm, *Os1900* expression clearly decreased after fertilization in WT and all tested mutant lines, while the rates of decrease were low in *M3*, *M6*, and *M9* (Supplementary Fig.11). At 30ppm, *M2* showed the same transcript level as WT (Fig. 5f, Supplementary Fig. 11). In addition, the expression of *M1*, *M4*, *M7* and *M8* have clear differences from those in WT under four fertilization conditions (Fig. 5f), suggesting that deletions with *M1*, *M4*, *M7* and *M8* contain critical cis-regulatory elements to determine the basic levels of *Os1900* gene expression. In contrast, *M3*, *M6*, and *M9* mutants with more gentle slopes than WT, exhibited less responsive to fertilizer at 30 ppm (Supplementary Fig.11; Supplementary Data 8), suggesting that these deleted regions contained regulatory elements needed for *Os1900* induction in response to nutrient deficiency. These results demonstrate that several novel deletion alleles of *Os1900* promoter regions

exhibited various fertilizer responses, which were different from those in WT.

Phosphate starvation responses (PHRs), which are required for mycorrhizal symbiosis, regulate symbiosis related genes via PHR1-binding sequences (P1BS) motif[16]. Thus, P1BS is a motif existing in promoter region of Pi starvation-responsive genes and distributed in both the *Os900* and *Os1900* promoter (Fig. 5d, e), which are covered in the deleted areas of *M2*, *M4*, *M7* and *M9* mutants (Fig. 5d). As shown in Fig. 5f, the transcript level of both *M4* and *M7* decreased under four fertilization conditions. However, *M2* was the only one line showing no different expression under all the tested conditions compared with WT (Fig. 5f). Besides, even one P1BS was found in *M9* deletion region, the expression of *Os1900* only showed difference at 50 ppm (Fig. 5f). These results suggest that not only the presence of the known key cis-regulatory elements, but also the positions of the deletions against the start sites of *Os1900* transcription could be critical for changing the distance between RNA polymerase complexes and the binding sites for the responsible TFs. Otherwise, some combinations of other cis-regulatory motif(s) with these P1BS would be involved. It is of note that there must be other critical cis-regulatory elements for nutrient deficiency since there are no P1BS in the *M3* and *M6* deleted regions. In addition, the one P1BS deletion in *M4* region caused the increase of tiller numbers due to low expression of *Os1900*, but was still able to respond to the nutrient deficiency. Thus, the role of P1BS sites in response to nutrient deficiency was not clear in *Os1900*.

CRE mutations likely represent genetic accelerators for many biological and agronomical traits during species diversification, domestication, and breeding processes[45]. As *os1900* mutants show increased tiller under some conditions (Fig. 4d), we further investigated tiller number dynamics in these promoter mutants under minor and trace fertilization conditions. Under minor conditions, *M3*, *M4*, and *M6* mutants produced more tillers than WT (Fig. 6a). It is especially of note that *M4* showed decreased *Os1900* expression under all the tested fertilization conditions (Fig. 5f). At trace fertilizer concentrations, *M1* and *M6* mutants exhibited slight decrease in tiller number compared to WT (Fig. 6b). All of them indicated a weak correlation between these tiller numbers and expression phenotypes. Crop yield was

affected by other factors in addition to tiller number. Indeed, seed fertility led to the decrease yield of *M3* and *M6* (Supplementary Fig. 12) although their tiller numbers were increased. Only *M4* was identified with significant increase so far in terms of grain yields under the laboratory cultivation conditions (Supplementary Fig. 12a). Although the evaluation of yield needs to be carried out in the paddy field in the future, we can speculate that such increases in mature tiller numbers may result in higher panicle numbers and yields in rice cultivation with a less fertilizer use.

Besides, any promoter mutational lines of agriculturally important quantitative trait genes can create novel useful alleles[43]. As an example, here, we demonstrated that *M4* mutant, an *Os1900* promoter deletion mutant, presented lower gene expression of *Os1900*, larger tiller numbers, and higher grain yields under minor fertilizer conditions. This may lay a foundation for breeding of rice cultivars adjusted to low fertilizer cultivation conditions in the future.

In this work, the genetic analysis of CRISPR/Cas9 mutants revealed that *Os1900*&*Os5100* are genetically responsible for the conversion of CL to CLA, which regulate tiller numbers. We reliably identified the role of *Os1900* as a fertilizer responsive gene under real paddy-field cultivation conditions and demonstrated that *Os1900* transcriptional regulation induced by fertilizers was important in controlling the tiller number for rice. Although further detailed CRE identification is still essential, our results clearly indicate that *Os1900* is a key gene involved in the regulation of tiller number by fertilization in rice cultivation, and that deletion mutations in the *Os1900* promoter region can maintain tiller numbers under minor fertilizer concentrations, which may contribute to the development of novel cultivars for sustainable agriculture.

## Methods

### Plant materials and growth conditions

*Oryza sativa* subsp. *japonica* cultivar Koshihikari plants grown in a paddy field were used for transcriptome analysis. All individuals were divided into two groups: normal transplantation group and delayed transplantation group. Each group was divided into three small groups: fertilization 0 time, 1time, and 2 times. After the first fertilization, we took the samples of rice leaves for 8 times at 10:00 am or 4:00 pm randomly of different dates. Practice amount of a typical commercial fertilizer for rice cultivation (~3 kg N/10a, ~4 kg P/10a, ~5 kg K/10a for both the first and the second fertilization) were used as indicated in instructions for use in this experiment.

The *MAX1*-like mutants, *os1900*&*os5100*, *os1900*, *os5100*, and *os900*&*os1400* were generated by CRISPR/Cas9 in the c.v. Koshihikari WT background. The Nipponbare cultivar was used as the background of *Os1900* promoter mutants.

Four distinct cultivation conditions of WT and *os1900*&*os5100* mutants, *os1900* promoter mutants for tiller number counting are (a) normal fertilization condition: 600 g fertilized soil/per plant; (b) minor fertilization condition: 300 g fertilized soil plus 300 g no-fertilized soil/per plant; (c) trace fertilization condition: 150 g fertilized soil plus 450 g no-fertilized soil/per plant; (d) no fertilization condition: 600 g no fertilized soil/per plant. The fertilized soil for rice seedling was Bonsol ($N = 1$ g, $P = 3$ g, and K = 1.7 g per 2.2 kg soil, Sumitomo Chemical CO., LTD.). The no fertilized soil was purchased from TACHIKAWA HEIWA NOUEN CO., LTD.

The four growth conditions for WT and *os1900*&*os5100* mutants were long day (14.5 h light/9.5 h dark) and low temperature (17–21 °C; 12 h light/12 h dark); short day (10 h light/14 h dark) and low temperature; long day and high temperature (26–30 °C; 12 h light/12 h dark); and short day and high temperature.

Kimura's B[46] solution was used as liquid fertilization, for *Os1900* expression of WT and *Os1900* promoter mutants. Liquid fertilizer is composed of N, P, K, and other related salts dissolved in ddH₂O. We used TDS (total dissolved solids) as a fertilizer status to measure the amounts of soluble ions in the aqueous solution. In order to exclude the effect of water absorbed by plants on fertilizer concentration, the solution volume was adjusted the same every time. In all cultivation experiments related to tiller number, "n" represents biological replicates.

### Field transcriptome analysis

Samples from three plants were pooled as single samples for transcriptome analysis. RNA was extracted using TRIZOL reagent (Invitrogen, Carlsbad, CA, USA). An Agilent 180k custom arrays were used for hybridization analysis. Gene expression data for each probe were processed using the Agilent protocol and transformed to $\log_2$ scale. Then, q-spline normalization[47] was applied to the distinct arrays. After normalization, transcriptome data were compared using paired *t*-tests, and false discovery rates and fold changes were calculated. All expression data and all tested sample combinations and probe sequences are presented in Supplementary Data 2.

### Real-time qPCR analysis

Fully emerged leaves or seedling bases (leaves removed) were used for all expression analyses. Leaves and shoots from three plants or seedling bases from five individuals were pooled as a single sample for real-time PCR, with three or four biological replicates. Total RNA was extracted using TRIZOL reagent (Invitrogen, Carlsbad, CA, USA), 2.4 μg of total RNA was used for cDNA synthesis with the ReverTra Ace qPCR RT master mix (Toyobo, Tokyo, Japan).

Real-time PCR was conducted using TaqMan Fast Universal PCR Master Mix (Applied Biosystems, Foster City, CA, USA) and the StepOne Real-Time PCR System (Applied Biosystems). The rice ubiquitin gene (*Os02g0161900*) was used for normalization. Gene-specific primers and TaqMan probe sequences are listed in Supplementary Data 9. To account for expression dynamics and data fluctuations, all gene expression data (or estimated copy numbers of target genes) were logarithmically transformed.

### RNA-seq analysis

Rice seedlings of WT and *os1900*&*os5100* mutants were grown under minor fertilization until 8-leaf age. For the first tiller base samples, we removed the outer old leaves and collected 2 cm parts from the base. For leaf samples, we used parts of the mature seventh leaf blade. RNA-seq library were prepared using TruSeq Stranded Total RNA Library Prep Plant Kit (Illumina, San Diego, CA, USA) and the RNA libraries were sequenced using Illumina NovaSeq6000 with paired-end 100-bp at Macrogen Japan. Adaptor sequences were trimmed using Trimmomatic version 0.39[48]. Trimmed reads were mapped to the IRGSP-1.0 genome assembly of rice using the STAR aligner version 2.7.10a[49]. For gene models of GTF file, we combined representative genes and computationally predicted genes in IRGSP-1.0 and removed rRNA and tRNA gene models. For leaf samples, we found that about 60 percent of reads were mapped to multiple loci, and that top over-represented sequences were mapped to plastid genome as well as nuclear genome. We discarded these reads and used remaining about 15 million reads mapped to unique loci for analysis. For first tiller base samples, about 10 to 20 percent of reads were mapped to multiple loci, and about 35 million reads mapped to unique loci were used for analysis. Read counts were quantified using RSEM version 1.3.3[50] and normalized using edgeR version 3.30.3[51] in R. Lowly expressed genes were filtered out using filterByExpr function and DEGs were extracted using glmQLFTest function in edgeR. Using the list of phytohormone related genes as described in previous reports[52–64] (Supplementary Data 10), gene expression was examined. Heatmaps were drawn using ggplot2 package in R[65]. The deposit IDs of our Illumina data for RNA-seq are: PRJNA889215.

Some key genes expression was confirmed with real-time PCR using same samples (Supplementary Fig. 7 and 10).

### CRISPR/Cas9 constructions and mutant genotyping
CRISPR/Cas9 (http://crispr.hzau.edu.cn/CRISPR2/) was used to generate *MAX1-like* mutants and *os1900* promoter mutants. gRNAs were ligated using the plasmid pU6gRNA. After PCR amplification, PCR products were assembled into pZH-OsU6gRNA-MMCas9 and transformed into *Escherichia coli* to obtain the target clone. The final target vector was subjected to *Agrobacterium tumefaciens*-mediated transformation. $T_0$ generation plants were hydroponically cultivated for 3 days; after the roots grew out, they were transplanted in soil under standard cultivation conditions in an artificial climate chamber. All gRNA sequences are listed in Supplementary Data 9. The *MAX1* mutants were genotyped by sanger sequencing in the $T_0$ generation. Promoter mutant genotyping was performed via PCR in the $T_0$ generation to select those with > 100 bp deletions, followed by Sanger sequencing in the $T_1$ generation.

### k-mer analysis
We obtained 12 Gbp Illumina data for the *os1900&os5100* mutants and WT. Paired-end fastq raw Illumina next-generation sequencing data were analyzed using k-mer software[66], and k was 20. $\log_{10}$ values of (k-mer for the double mutants/k-mer for WT) were plotted. Reference sequence data (Nipponbare) for five *MAX1-like* genes was obtained using the RAP-database (IRGSP1.0).

### In situ hybridization
WT plants were grown in unfertilized soil. At 17 days after germination, one shoot apex from each seedling was sampled before and after fertilization, dissected, fixed in 4% paraformaldehyde in 0.1 M sodium phosphate buffer for 48 h at 4 °C, and then dehydrated in a graded ethanol series. The ethanol was replaced with Histo-Clear (National Diagnostics, Atlanta, GA), and the samples were embedded in Paraplast Plus (Leica, Wetzlar, Germany). Paraffin sections (thickness, 8 μm) were placed on microscope slides coated with 3-aminopropyl triethoxysilane (Matsunami Glass, Osaka, Japan). To generate a probe for *Os1900*, the full-length cDNA clone AK122077.1 was obtained from the National Institute of Agrobiological Sciences (NIAS) Genebank (http://www.dna.affrc.go.jp/distribution/) and used as a template. For *Os5100*, a cDNA fragment was amplified by PCR using the primer set 5′-ATC ACATACGAGTGATCAAAAGCTC-3′ and 5′-TGAGCACCATCCCGAACT G-3′, and cloned into the pCR Blunt II TOPO vector (Invitrogen). The cDNA of *Os1900* and *Os5100* was linearized using AvaI (New England Biolabs) and PvuII (Takara, Shiga, Japan), respectively, and then digoxigenin-labeled antisense riboprobes were transcribed using T3 (Roche) or T7 (Takara, Shiga, Japan) RNA polymerase with DIG-RNA labeling mix (Roche). For *H4*, full length cDNA of H4 was used for transcribing digoxigenin-labeled antisense riboprobe. In situ hybridization and immunological detection of the hybridization signals were performed as previously described[67].

### LC-MS/MS analysis of endogenous 4DO, CL, CLA and MeCLA
Rice plants were grown as previously described[68], with minor modifications. Eleven-day-old seedlings grown on 0.6% agar media of hydroponic nutrients under a 16 h light/8 h dark photoperiod were transferred to glass vials containing 50 mL hydroponic nutrient solution without inorganic phosphate to activate SL biosynthesis[25]. The shoot and root of the 32-day-old seedlings were harvested and homogenized in 12.5 mL of acetone containing stable isotope-labeled internal standards [[6′-$D_1$]-4DO[11], [1-$^{13}CH_3$]-($S$)-CL[69], [1-$^{13}CH_3$]-*rac*-CLA[22], [10-$D_1$]-*rac*-MeCLA[22] using POLYTRON PT3100D (Kinematica).

The filtrates were evaporated under nitrogen gas until acetone was almost removed. Then, 1 mL of distilled water and 2 mL of saturated NaCl were added, and the concentrated filtrates were extracted with 5 mL of ethyl acetate (EtOAc) three times. The organic phase was then evaporated to dryness under nitrogen gas, dissolved in 1 mL of acetonitrile, and purified using the Bond Elut DEA (100 mg) 1 mL cartridge (Agilent) (SPE method A, see Supplementary Data 11-I). The Fraction A after the SPE method A was further separated into CL-, MeCLA-, and 4DO-containing fractions using the Sep-Pak silica 1 cc vac cartridge (Waters) (SPE method B). The CL- and MeCLA-containing fractions were then purified using the Sep-Pak cyanopropyl 1 cc vac cartridge (SPE methods C and D, respectively). CLA in Fraction B after the SPE method A was purified using the Sep-Pak silica 1 cc vac cartridge (Waters) (SPE method E). The eluates were evaporated to dryness, dissolved with acetonitrile, and subjected to LC-MS/MS. It is noted that the filtrates of the acetone extract were directly used to analyze CL levels in shoots and roots of the *os900&os5100* double mutant. LC-MS/MS analysis was carried out by using an X500R QTOF system (AB SCIEX). LC conditions and MS methods are shown in Supplementary Data 11-II, -III, respectively. Data acquisition and quantitative processing were performed by using the SCIEX OS software (version 2.0.1) (AB SCIEX). Calibration curves generated using labeled and non-labeled compounds were used for quantification of SLs.

### Bioassay of (+)-GR24 on rice seedlings
The (+)-GR24 treatment was performed as previously described[68], with minor modifications. One-week-old rice seedlings grown on 0.6% agar media of hydroponic nutrient under a 16 h light/8 h dark photoperiod were transferred to glass vials containing 13 mL hydroponic nutrient solution. All hydroponic nutrient media contained 5 mM MES and were adjusted to pH 5.7. Dimethyl sulfoxide (DMSO) was used to dissolve (+)-GR24, and DMSO (mock) or (+)-GR24 was added to the hydroponic nutrient solution for a final DMSO concentration of 0.1% (v/v). The rice seedlings were incubated under the same conditions for 8 days; the hydroponic nutrient solution was supplemented after 4 days of incubation. The first and second tillers were measured using a digital caliper under a microscope.

### Tiller number counts
Tiller numbers of tested rice seedlings were counted every 3 days. Then GLMs or GLMMs (including full model and reduced models) to explain the tiller number increase were made using the presence of *Os1900&Os5100* and growth stages as parameters using lme4 in R and then ANOVA analysis of the models were performed to get *P* values for the effects of *Os1900&Os5100* affecting tiller numbers.

## Data availability
The data that support the findings of this study are available from the corresponding author (TI) upon request. RNA-seq data associated with this study have been deposited at NCBI under accession code PRJNA889215. Source data are provided with this paper.

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

## Acknowledgements

We would like to thank Drs. Masaki Endo and Seiichi Toki (National Agriculture and Food Research Organization) for providing CRISPR/Cas9 vectors. We also thank Dr. Kenichiro Hibara (currently, Kibi International University) for teaching some key protocols to J. Cui at the beginning in this study and Dr. Mingchao Wang (Dresden University of Technology) for proofreading some languages. We are grateful to the System for Development and Assessment of Sustainable Humanosphere (RISH, Kyoto University) for providing a greenhouse to KM and SY. This work was supported by JSPS KAKENHI under grant number JP17H06246 (to T.I.), JP22H05180 (to T.I.), JP22H0517222 (to T.I.), JP22H00367 (to T.I.), JP19H02892 (to K.M.), JP17H06474 (to S.Y.), by the Human Frontier Science Program Organization under grant number RGP0011/2019 (to T.I.), and by Cabinet Office, Government of Japan, Cross-ministerial Moonshot Agriculture, Forestry and Fisheries Research and Development Program, "Technologies for Smart Bio-industry and Agriculture"(funding agency: Bio-oriented Technology Research Advancement Institution) (JPJ009237 to T.I.). the Collaborative Research Program of Institute for Chemical Research, Kyoto University (grant # 2020-92) (to T.I., K.M., and S.Y.)

## Author contributions

T.I. conceived the original idea on this work, organized all the experiments and mainly revised the manuscript. T.I. and N.N. performed the field transcriptome analysis. With substantial support by K.K. and K.S., N.N. made the *os1900* & *os5100* and *os900* & *os1400* double mutants and check the tiller number phenotypes firstly. K.M. and S.Y. performed SL analysis and GR24 application assay. M.M. and J.I. performed in situ hybridization analysis and RNA-seq analysis. J.C. performed all of the rest experiments including related to genotyping and phenotyping of all mutants, the entire experiments on the promoter deletion mutant analysis and wrote the original manuscript. All the authors revised the manuscript.

## Competing interests

The authors declare no competing interests.
