## [Peer Review File · Nature Communications]

Fertilization controls tiller numbers via transcriptional regulation of a MAX1-like gene in rice cultivationReviewer #1 (Remarks to the Author):

The manuscript from Cui et al. presents a large set of results, representing a huge quantity of work, that could be divided into three parts/experiments and from my point of view, could be presented and discussed in more details in 3 different manuscripts. The objectives of the study are not clearly explained in the introduction. In the course of the manuscript, we understand that it is to first identify genes responding to fertilization in field conditions then to elucidate the mechanism by which fertilization controls tiller numbers and finally to identify the important cis-regulatory sites in the Os1900 promoter regulating its transcriptional response to fertilization.

A large transcriptomics experiment was first performed from leaf samples of rice grown in field conditions with normal and delayed transplantations and under 3 fertilization protocols. Leaf samples were harvested at various times/developmental stages/environmental conditions (Supplem Fig 1). From this experiment, the authors used methods and statistics (not very clear to me) to identify genes responding to the fertiliser factor only. A list of «107 genes as stable fertilizer response genes in rice » is given without further discussion. They decided to focus on Os1900, «a remarkable gene » because of its regular transcriptional response to fertilizer (from what I understood) and because this gene, homolog to MAX1 in Arabidopsis, is known to be involved in strigolactone (SL) biosynthesis, which is known to be strongly induced in response to P (and in some species, to N) starvation. In another set of experiments, in laboratory conditions, they show that Os1900 is indeed induced in response to a decrease in fertilizer.

In a second part, which contains data of interest, CRISPR-Cas9 single and double mutants are obtained for the 4 rice MAX1 homologs. Several studies on the rice MAX1 homologs have already been published (in particular for investigating their biochemical function) but mutants have never been produced and characterized. Tillering in different nutrient conditions and SL quantifications in both shoot (which is remarkable !) and root were analyzed in the double mutants. It would have been of interest to have more comparisons between the two double mutants as it is quite interesting to observe the mild phenotype of the os900 os1400 double mutant in comparison to the os1900 os5100 double mutant. Again the objective of this part is not clear as many questions of interest are addressed such as the identification of the SL acting as the branching inhibitor, the transport/synthesis of SL between root and shoot in rice but conclusions are not well highlighted. In a third part, a series of CRISPR-Cas9 deletion alleles covering the Os1900 promoter region were produced to identify the regions responsible for associating fertilization with tiller number control. Transcript levels of Os1900 were analyzed in WT and in the mutants which displayed different fertilizer responses from WT. But no clear results can be provided from these experiments. Moreover many comments could be given about the logic of some sentences which is not clear, the organization of the manuscript or the methods used (e.g. explain better what is the total dissolved solids, TDS used for fertilization...)

Reviewer #2 (Remarks to the Author):

Overuse of chemical fertilizers has led to aquatic and terrestrial pollution, including water eutrophication and soil consolidation. To achieve sustainable agriculture, it is urgent to reduce global chemical fertilizer use. Fertilization results in biological changes in many aspects of plant growth, especially promote elevation of tiller number. Recently, scientists have made several important progresses in the molecular mechanisms underlying this regulation, such as regulation of tiller number by different levels of nitrogen.

In the manuscript, the authors discover a novel regulatory mechanism of fertilization on tiller number through analyzing the effects fertilization of on Os1900 expression. They systematically conducted experiments to elucidate the relationship between the tiller number control and transcriptional regulation of Os1900 under different conditions of fertilization. Genetic analysis using a series of deletion mutations on Os1900 promoter suggest that Os1900 promoter mutations may be applied in breeding program. The research topic is important and of general interest. But the causal roles and importance of regulation on Os1900 expression in fertilization controlled rice tillering still need further elucidation.

The following are points need to be further addressed:

1. Figure 1c showed that the transcription of Os1900 under fertilization condition is higher than that without fertilization at natural field samples. However, Figure 1f showed that fertilization repressed transcription of Os1900 at lab condition. The phenomena are rather confusing. What's the reason of this difference.
2. In lines 281-283, the authors mentioned that "Since 4DO was detected in both shoots and roots of the os1900&os5100 double mutants more than in those of WT, 4DO is not a major SL product in inhibiting tiller elongation in rice". What's the experimental evidence for this conclusion? If 4DO is not a major SL product in inhibiting tiller elongation in rice, which molecular should be the major SL product in rice to inhibit tiller elongation?
3. The authors should give more evidence to investigate the content of CL and CLA in shoot and root of WT and os1900&os5100 double mutant. If they indeed catalyze the reaction of CL to CLA, why the CLA levels significantly increased in root tissues of the os1900&os5100 double mutant. If this is related to the expression levels and tissue specificity of Os900 and Os1400, more genetic evidence should be provided.
4. Results in Figure 5 and 6 indicate that a series of deletion alleles Os900 is related to different fertilization response, and also found several predicted cis-regulatory elements (CREs) in these deletions. This is an interesting point and needs further investigation. The authors are suggested to identify the importance of these CREs in fertilizer responses through genetic analysis and biochemical assays. For instance, to generate CRISPR/CAS9 mutants defective in specific CREs and detected their responses to fertilizer. Or they should identify the transcription factors that regulate expression of Os1900 and other SL biosynthesis genes through these CREs. Otherwise, the importance of these CREs and also the regulation on Os1900 expression in fertilization controlled rice tillering is not clear.

Minor points

5. As showed in Figure 3b, Figure 3c and Supplementary Figure 5a, the error bars were too big. The reviewer strongly suggested the authors to repeat these experiments.
6. In line 265, Os1900 should be Os1400.
7. In line 243, Fig. 3a should be Fig. 3a-b

Reviewer #3 (Remarks to the Author):

Summary:

The authors conducted transcriptomic analyses on rice plants grown under various different fertilizer treatments to identify key genes respond to fertilizer, independently of changing environmental conditions. They identified that Os1900, a MAX1-like gene involved in the strigolactone synthesis pathway, strongly responds to nutrient deficiency. By generating mutants in this gene and the closely related Os5100, they showed that both these genes are required to control rice tiller number. Tissue-specific expression of Os1900 in shoots alters SL-intermediates distribution compared to roots, indicating a possible way of controlling tissue-specific responses. Finally, the authors generate a series of mutants in the Os1900 promoter, producing rice plants with potential agronomical properties.

Comments to authors:

1- I find that the introduction section is lacking key references describing previous work that studies strigolactone involvement in plant development in response to nutrient deficiency. The authors discuss the role of SLs in indirect nutrient uptake via with arbuscular mycorrhizae symbiosis, but there is ample work showing how strigolactone-related genes are regulated in response to nutrient deficiency and more importantly, how they affect branching (eg., (Sun et al., 2014; Marzec et al., 2013; de Jong et al., 2014)). Moreover, it is stated in lines 59-63 that "Fertilization results in biological changes in many plant growth parameters (...). However, the mechanisms by which fertilization controls these traits remain largely unknown", while it is obviously clear that there is ample literature on this matter, with whole agronomy research fields dedicated to this matter. Please rephrase this sentence.

2- Lines 120-122 and Figure 1c. A single Fold Change value is being reported. According to the legend, it seems like averaged fold change of Fertilized vs non-fertilized at each time point is being represented, but it is not sufficiently clear. Also, there are one/two fertilization samples, are both treatments also being averaged? Please clarify more in detail in the text/methods section.

3- Lines 146-149. Although it might not be the scope of this paper, probably there is additional interesting information to be extracted from the transcriptomic analysis. Are there any other interesting fertilization-responding genes? At least, the response of other MAX1-like genes should be included in the text, even if they do not respond in the collected tissue (this would more strongly highlight the importance of Os1900 response in this tissue).

4- Lines 160-162. The authors state that they identified the Os1900 response "under normal rice cultivation conditions" but they measured its response comparing fertilized to non-fertilized conditions. Could the authors provide a rough estimate of P/N/K concentration in the soil from which these plants were grown? Probably P/N/K concentration was low in non-fertilized samples explaining why Os1900 was induced in field conditions.

5- Lines 163-171 and Figure 2d. The authors describe that only the Os1900 Os5100 double mutant shows a tillering phenotype. It is surprising that the single Os1900 mutant has a wild-type tillering phenotype. If Os1900 is induced under nutrient deficiency conditions, one would expect that the single Os1900 mutant would show a significant phenotype in this condition. The authors show in Supplementary Figure 3a that this is not the case.

These results suggest that maybe there is a compensatory mechanism by which Os5100 is induced in the absence of Os1900. I suggest that the authors measure Os5100 mRNA levels (and maybe other MAX1-like genes) in single Os1900 mutants and vice-versa, under normal and nutrient deficiency conditions to clarify this question.

I would also suggest to include a quantification of the tillering phenotype shown in supplemental figure 3. While the image might be representative, only a single picture is shown.

6- Lines 242-243. The authors state that "Significant CL accumulation was observed in os1900&os5100 double mutants in both shoots and roots" while the results show high variability in the quantification of CL in roots and CLA in shoots, with p-values showing low statistical significance. If the authors want to show statistically significant changes, I would suggest increasing n. Alternatively, I would suggest to rephrase the text.

7- Lines 289-290 "Tiller elongation is inhibited at axial bud regions by SLs; however, it remains unknown whether SLs can migrate to action sites." I find it quite surprising that the authors make this statement. In fact, intermediate products of SLs were described as mobile signals before SL structure, synthesis and signaling were known (Booker et al., 2005). More specifically, MAX1 was identified as a key gene able to process the SL intermediate signal in axillary buds to produce a branch inhibiting SL.

8- Lines 361-363. In the experiments with Os1900 promoter mutant plants, the authors state that "Rates of decrease were low in M3, M6, and M9". These lower rates of decrease could be also partially explained by lower "starting" expression in 30 ppm 0h conditions (although other mutants still show a "WT" decrease in Os1900 levels). In the text it is stated that "M3, M6 and M9 contain critical cis-regulatory elements for fertilizer responses" while it seems to be the contrary, they seem to contain regulatory elements needed for Os1900 induction in response to nutrient deficiency. Moreover, they do not seem to be "critical" as there is still some degree of response. In fact, it seems that mutants with lower Os1900 induction at 30 ppm 0h show the strongest phenotype in non-fertilized conditions, indicating that Os1900 levels correlate directly with the tiller number. Could the authors represent Os1900 expression levels with tiller number? Is there a direct relationship? As the authors suggest, this might not be the case (as M4, M6, M7, M8, and M9 mutants produce more tillers than WT), but I think it would be interesting to show the data to back up this conclusion.

Also, in lines 368 to 388, conserved CREs in other SL genes are studied, while these genes show clearly different expression patterns (shoot vs root) and different responses to nutrients. To me, the conservation of these motifs and their location in the promoter does not add any meaningful

information. Perhaps only the closest MAX1-like genes should be included in this analysis.

9- The authors suggest that the increased tiller number in some Os1900 promoter mutants “may result in higher panicle numbers and yields in rice cultivation”. I think this is a key missing result. Could the authors quantify panicle number/yield of these mutants to show if this is true?. Also, although Os1900 seems to be mainly expressed in shoots, it would be important to show if those mutants have any effect on AM symbiosis. Is the change in tiller number just a consequence of different strigolactone processing in shoots or is indirect nutrient uptake involved?

Minor comments:

1- I suggest including the treatment diagram shown Supplementary figure 1a into the main figure 1. It would make it much easier for the reader to comprehend the sampling procedure. Also, explaining it in more detail in the manuscript would make it easier to understand.

2- Please change the colors of Supplementary Figure 1d. In its current state it is almost impossible to follow the legend, as there are duplicated colors/shapes. One idea could be using empty/filled shapes for morning and afternoon samples, if indeed they PCA splits the samples between these factors.

3- Please change order of figure 3. Panel h should be panel a, and cited in lines 220-226. Also, figures 3 and 4 are not cited in order in the text, making it harder to follow.

4- Lines 205-206 and Supp figure 4. Please explain more in detail the growth conditions of this experiment. In the text it is stated that the authors “compared the effects of fertilization” but it seems that plants were grown only under one condition. Were they fertilized? Were they under nutrient deficiency?

5- In Figure 4b, I assume the legend refers to 1h of fertilization, but it is not stated.

6- Throughout the whole paper tiller number phenotyping is conducted with what seems to be a very low population. In the figures where n is reported it varies from 4 to 6, and in some cases, it seems that n equals 3. Are these individual plants? Are these averaged experiments of larger populations? Please clarify in the figure legends and/or methods section.

References:

- Booker, J., Sieberer, T., Wright, W., Williamson, L., Willett, B., Stirnberg, P., Turnbull, C., Srinivasan, M., Goddard, P., and Leyser, O. (2005). MAX1 encodes a cytochrome P450 family member that acts downstream of MAX3/4 to produce a carotenoid-derived branch-inhibiting hormone. *Dev. Cell* 8: 443–449.
- de Jong, M., George, G., Ongaro, V., Williamson, L., Willetts, B., Ljung, K., McCulloch, H., and Leyser, O. (2014). Auxin and strigolactone signaling are required for modulation of Arabidopsis shoot branching by N supply. *Plant Physiol.* 166: 384–395.
- Marzec, M., Muszynska, A., and Gruszka, D. (2013). The Role of Strigolactones in Nutrient-Stress Responses in Plants. *Int. J. Mol. Sci.* 2013, Vol. 14, Pages 9286-9304 14: 9286–9304.
- Sun, H., Tao, J., Liu, S., Huang, S., Chen, S., Xie, X., Yoneyama, K., Zhang, Y., and Xu, G. (2014). Strigolactones are involved in phosphate- and nitrate-deficiency-induced root development and auxin transport in rice. *J. Exp. Bot.* 65: 6735–6746.

The point-by-point responses to the reviewers' comments with the reports reproduced verbatim

Manuscript ID: NCOMMS-22-05588A

Title: Fertilization controls tiller numbers via transcriptional regulation of a *MAX1*-like gene in rice cultivation

To all the reviewers,

Thanks for taking care of our revised manuscript.

First of all, in this revised manuscript, we would like to tell you that as a result in an additional experiment we demonstrated that both grain yields and tiller numbers significantly increased in a few promoter deletion mutants in *Os1900* under a less fertilizer condition, clearly suggesting that these *Os1900* promoter deletion mutants can be agriculturally useful (Supplementary Fig.12). These time-course changes in tiller number were also used to make GLM or GLMM models, which were selected by AIC in this experiment and obtained significant P-values as shown in Fig.2d, Fig.6, and Supplementary Fig.4. We are sure that the data become more reliable statistically. These results clearly demonstrated that some *Os1900* promoter mutations have great potential for the use of breeding programs towards sustainable rice production. We highlighted all the changes in purple color in this revised manuscript and Supporting Information.

In addition, as a solution to the editor's comment, we first examined the expression of these *MAX1-like* genes, i.e., *Os900*, *Os1400* (known for converting CL into CLA), *Os1900*, and *Os5100*, in shoots and roots of WT and *os1900&os5100* double mutants by using samples obtained in the same manner for SLs biochemical analysis (shown in Fig. 3d,e). Then, we found that the amount of *Os900* & *Os1400* transcripts were not affected by the *os1900* & *os5100* double mutations. Thus, we can conclude that the accumulation of CL in the *os1900* & *os5100* double mutants was due to loss of enzymatic activity of *Os1900* & *Os5100* gene product.

In terms of increasing reproducibility of our data, as also suggested by other reviewers' comments, we next performed SLs biochemical analysis independently again to determine the role of *Os1900* and *Os5100* more reliably in controlling CL-to-CLA conversion *in planta*. This time, we increased the biological replicate number to five. In the new experimental results, there was a large accumulation of CL in the shoots and roots of the *os1900* & *os5100* double mutants, reproducibly, and CLA could hardly be detected as shown in Fig.3b, c.

To determine the genome-wide effects of *Os1900* and *Os5100* as the result of CL-to-CLA conversion *in planta*, we further performed a series of RNA-seq analyses to compare the expression of SL related genes, other phytohormone related genes, and bud dormancy related genes in leaves and tiller bases in rice. The RNA-seq results and related RT-qPCR data are presented in Supplementary Figs. 7-10. We have obtained many novel findings related to the role of *Os1900* & *Os5100*, suggesting the central roles of *Os1900* & *Os5100* for gene networks related to various phytohormone biosynthesis and signaling pathways, not only SLs, but also GAs, ABA, JAs, Auxins, and CKs. In addition, *FT*-like genes related to bud dormancy would be affected in the *os1900* & *os5100* double mutants (Supplementary Fig. 9b), revealing a more critical role of *Os1900* & *Os5100* than those found in our previous manuscript.

Reviewer #1:

The manuscript from Cui et al. presents a large set of results, representing a huge quantity of work, that could be divided into three parts/experiments and from my point of view, could be presented and discussed in more details in 3 different manuscripts. The objectives of the study are not clearly explained in the introduction. In the course of the manuscript, we understand that it is to first identify genes responding to fertilization in field conditions then to elucidate the mechanism by which fertilization lines8controls tiller numbers and finally to identify the important cis-regulatory sites in the Os1900 promoter regulating its transcriptional response to fertilization.

Response: Thanks for this comment. Accordingly, we have changed the introductory

section to make the purpose of this work clearer, line 4 54-59; lines 82-93; lines 96-102. Our purpose in this work is to understand the responsible molecular mechanisms to control tiller numbers in response to fertilizer in rice under natural paddy field conditions. Then, in this work, we clearly demonstrated that transcriptional regulation of *Os1900* gene is a key to control tiller numbers in rice.

In this revised manuscript, to make this work more unified, we newly obtained the tiller number of *os1900* single mutant under minor- and trace-fertilization conditions (Fig. 4c, d) to provide a better link among our discoveries. To highlight the core roles of SL network in affecting various other phytohormone functions, we have demonstrated various novel molecular biological roles of *Os1900* & *Os5100* in the revised manuscript by RNA-seq analysis (Supplementary Fig. 7-11). In addition, we have moved the original Fig. 1e, f to the current Fig. 4a, b for a better understanding of readers. To link the possible three parts more logically, we have put several sentences to explain how new parts are related to previous parts, lines 146-148 (between first and second parts); lines 399-401 (between second and three parts).

A large transcriptomics experiment was first performed from leaf samples of rice grown in field conditions with normal and delayed transplantations and under 3 fertilization protocols. Leaf samples were harvested at various times/developmental stages/environmental conditions (Supplem Fig 1). From this experiment, the authors used methods and statistics (not very clear to me) to identify genes responding to the fertiliser factor only. A list of «107 genes as stable fertilizer response genes in rice » is given without further discussion. They decided to focus on Os1900, «a remarkable gene » because of its regular transcriptional response to fertilizer (from what I understood) and because this gene, homolog to MAX1 in Arabidopsis, is known to be involved in strigolactone (SL) biosynthesis, which is known to be strongly induced in response to P (and in some species, to N) starvation. In another set of experiments, in laboratory conditions, they show that Os1900 is indeed induced in response to a decrease in fertilizer.

Response: Thanks for this summary of the first part of our manuscript. Although we are starting a deep functional analysis of other fertilizer response genes than *Os1900* in

the list, we would like to share this information with other researchers in this field since we are apparently not able to work on all the genes in the list at once.

In a second part, which contains data of interest, CRISPR-Cas9 single and double mutants are obtained for the 4 rice MAX1 homologs. Several studies on the rice MAX1 homologs have already been published (in particular for investigating their biochemical function) but mutants have never been produced and characterized. Tillering in different nutrient conditions and SL quantifications in both shoot (which is remarkable !) and root were analyzed in the double mutants. It would have been of interest to have more comparisons between the two double mutants as it is quite interesting to observe the mild phenotype of the os900 os1400 double mutant in comparison to the os1900 os5100 double mutant. Again the objective of this part is not clear as many questions of interest are addressed such as the identification of the SL acting as the branching inhibitor, the transport/synthesis of SL between root and shoot in rice but conclusions are not well highlighted.

Response: Thanks for this summary of the second part of our manuscript. We agree that there are many interesting questions including the difference of actions between *Os1900&Os5100* and *Os900&Os1400*. Among them, we decided to focus on an important role of the transcriptional regulation of *Os1900* by fertilizer to control tiller numbers in rice. Thus, we set up the third part experiment.

In the second part, to be sure that *Os1900* & *Os5100* genes are key genes in this work, as a reply to the editor's comments, we newly performed RNA-seq analysis to reveal the novel gene network surrounding *Os1900* & *Os5100* genes, in addition to related gene networks in SL biosynthesis and signaling. Then, we have succeeded to address more questions and revealed various novel molecular biological roles of *Os1900* & *Os5100*, suggesting core roles of SL signals affecting various other phytohormone functions.

In a third part, a series of CRISPR-Cas9 deletion alleles covering the Os1900 promoter region were produced to identify the regions responsible for associating fertilization with tiller number control. Transcript levels of Os1900 were analyzed in WT and in the mutants which displayed different fertilizer responses from WT. But no clear results can be provided from these experiments.

Response: Thanks for this summary of the third part of our manuscript. We have revised this part as follows.

First, we re-scanned the CREs of the *Os1900* promoter region by focusing on P1BS (PHR1-binding sequence), which exists in the promoter of Pi starvation-responsive structural genes (Fig.5d). Secondly, to confirm the repeatability of tiller number in promoter mutants, we replanted the *M1-M9* mutants by increasing plant numbers to 8. We employed a more rigorous statistical method such as GLM or GLMM to calculate the p-value, and then investigated the grain yield of *M1-M9* mutant under laboratory conditions (Fig.6). Finally, we found that a few deletion mutants exhibiting more tiller numbers than those in WT under minor-fertilization cultivation conditions. Especially, the *M4* mutant showed more tillers and more grain yields than those in WT (the deleted region in *M4* contains one P1BS site) (Supplementary Fig.12).

Taken all together, in this work, we clearly show that transcriptional regulation of *Os1900* controls tiller numbers. The *M4* mutant phenotypes under the minor fertilizer conditions will lay the foundation for breeding novel rice varieties with low fertilizer demand and high yield.

Moreover many comments could be given about the logic of some sentences which is not clear, the organization of the manuscript or the methods used (e.g. explain better what is the total dissolved solids, TDS used for fertilization...)

Response: Thanks for your comments, we have carefully revised the manuscript.

About TDS:

To measure the gene expression shown in Fig. 4a, b, we used liquid fertilizer (Kimura's B) to cultivate plants. Liquid fertilizer is composed of N, P, K and other elements as salts dissolved in ddH₂O. We used TDS (total dissolved solids) as a fertilizer status to measure the amounts of ions in the aqueous solution. In order to exclude the effect of water absorbed by plants in changing fertilizer concentration, the solution volume was adjusted the same every time. This has been added to Methods, lines 552-555.

Reviewer #2

[Omitted] The research topic is important and of general interest. But the causal roles and importance of regulation on Os1900 expression in fertilization controlled rice tillering still need further elucidation.

Response: Thanks for your valuable comment. We agree that it was unclear how important is fertilizer- induced transcription of *Os1900* in the control of tiller numbers in rice in the previous manuscript. In this revised manuscript, we have largely improved the organization of the presentation and the data reliability of key findings such as *os1900* phenotypes under trace-fertilizer conditions, promoter mutant phenotypes, SL quantification, and the grain yield test. Thus, now it becomes easier to understand that *Os1900* transcriptional regulation is a key to control tiller numbers in rice. Furthermore, with the RNA-seq data, we demonstrate a more important biological role of *Os1900* with the support of *Os5100*, that is a central (or core) role to control most of the other phytohormone signaling networks.

The following are points need to be further addressed:

Comment 1: *Figure 1c showed that the transcription of Os1900 under fertilization condition is higher than that without fertilization at natural field samples. However, Figure 1f showed that fertilization repressed transcription of Os1900 at lab condition. The phenomena are rather confusing. What's the reason of this difference?*

Response: We appreciate this comment a lot for pointing this out. In fact, in both paddy field and laboratory conditions, the expression of *Os1900* was induced under fertilizer-starvation conditions, which is consistent with previous reports. We are sorry for a mistake in the caption in the previous Fig. 1, which has been corrected, lines 133-135. The FCs indicate the absolute values in Fig.1.

Comment 2. *In lines 281-283, the authors mentioned that "Since 4DO was detected in both shoots and roots of the os1900&os5100 double mutants more than in those of WT, 4DO is not a major SL product in inhibiting tiller elongation in rice". What's the experimental evidence for this conclusion? If 4DO is not a major SL product in inhibiting tiller elongation in rice, which molecular should be the major SL product in*

rice to inhibit tiller elongation?

Response: Thanks for this comment. Initially, we simply thought that 4DO is irrelevant to the tillering because of the more accumulation of 4DO in *os1900&os5100* than that in WT. In addition, Ito et al. (2022) in "canonical strigolactones are not the major determinant of tillering but rhizospheric signals in rice." tells us that since Os900 catalyzes 4DO synthesis in rice, *os900* single mutant can't produce 4DO; but *os900* mutant did not show more tillers. Thus, 4DO seemed not to be the main SLs to control tiller numbers. According to this paper, however, neither WT nor *os900* mutant, 4DO was detected in shoot. Thus, we think whether 4DO is tillering-inhibitory hormone still remains an open question at this moments. Thus, in this revised manuscript, we removed the statement that 4DO is not the main substance that controls tillering in rice. The corresponding part is in line 290-292.

Comment 3. The authors should give more evidence to investigate the content of CL and CLA in shoot and root of WT and os1900&os5100 double mutant. If they indeed catalyze the reaction of CL to CLA, why the CLA levels significantly increased in root tissues of the os1900&os5100 double mutant. If this is related to the expression levels and tissue specificity of Os900 and Os1400, more genetic evidence should be provided.

Response: Thanks for this valuable comment. We agree that the fluctuation in CLA level can easily lead to some confusions due to the big bar in our previous manuscript.

We thus performed SLs analysis again by increasing the replicate number to five. This time, compared with WT, CL largely accumulated in *os1900* & *os5100* mutants in both shoots and roots (Fig. 3b, c). And the amount of CLA was significantly lower in *os1900* & *os5100* mutants than that in WT in shoot and root (Fig. 3b, c), thus these data made us to lead simpler and clearer conclusions in this revised manuscript.

In addition, we examined the expression of *Os900*, *Os1400*, *Os1900* and *O5100* in shoots and roots using the same batches of samples for the biochemical analysis (Fig. 3d, e). For *Os900* and *Os1400*, no difference was observed between *os1900&os5100* double mutants and wild-type. Then we analyzed *Os900* and *Os1400* expression in stem bases and tiller bases. The results showed that the expression levels of *Os900* and *Os1400* were kept the same (or higher) in the *os1900&os5100* double mutant as (than) in the wild type (Supplementary 6 c, d). In addition, *Os1900* and *Os5100* have high

expression in both roots and shoots of WT; their expression decreased in *os1900* & *os5100* (Fig. 3d, e). Taken all together, the drastic CL accumulation in the *os1900* & *os5100* is likely to be due to loss of Os1900 & Os5100 enzyme activity, but not through the *Os900* & *Os1400*.

Moreover, we further performed RNA-SEQ transcriptome analysis of the *os1900* & *os5100* double mutants and the wild-type, and demonstrated that both *Os1900* and *Os5100* have a core role to control biosynthesis & inactivation of CK, ABA, Auxin and JA and/or their signaling pathways, subsequently controlling tiller elongation (or dormancy) (Supplementary Fig.7-10).

Comment 4. Results in Figure 5 and 6 indicate that a series of deletion alleles Os900 is related to different fertilization response, and also found several predicted cis-regulatory elements (CREs) in these deletions. This is an interesting point and needs further investigation. The authors are suggested to identify the importance of these CREs in fertilizer responses through genetic analysis and biochemical assays. For instance, to generate CRISPR/CAS9 mutants defective in specific CREs and detected their responses to fertilizer. Or they should identify the transcription factors that regulate expression of Os1900 and other SL biosynthesis genes through these CREs. Otherwise, the importance of these CREs and also the regulation on Os1900 expression in fertilization controlled rice tillering is not clear.

Response: We agree that generating CRISPR/Cas9 mutants defective in specific CREs and screening real CREs are good ideas, and now are doing in our laboratory. However, it takes over a year time to make CRISPR/Cas9 mutants, reconfirm the genotype in T1 generation, collect seeds, and conduct experiments using T2 generation plants in rice, and the expected process to reveal the entire cis elements in *Os1900* would be very complex (multiple CREs sites need to be knocked out, one by one), meanwhile the information on such deletion would be useful to make new rice cultivars adjusted to low fertilizer conditions. Thus, we decided to submit the data of deletions at this point. In this revised manuscript, we scanned CREs in the *Os1900* promoter region again and focused on the P1BS motif (the only known CRE for the Pi starvation response, which can interact with two TFs, *PHR1* and *PHL1*). In total, four P1BSs were found in *Os1900* promoter region (Fig. 5d). In addition, deletion region of *M4* mutant contained one P1BS. Under the four fertilization conditions (50ppm vs 245ppm and 30ppm vs

245ppm), the *Os1900* expression decreased in *M4* mutant (Fig.5f). Under minor fertilizer conditions, *M4* has increased tillers and grain yields (Fig. 6 and Supplementary Fig. 12). Thus, we believe that the *M4* deletion region contains a critical region regulating the transcription of *Os1900* in response to fertilizer.

Minor points

Comment 5. As showed in Figure 3b, Figure 3c and Supplementary Figure 5a, the error bars were too big. The reviewer strongly suggested the authors to repeat these experimnts.

Response: Thank you for your advice, we have repeated these experiments by increasing biological replicates to five (Fig. 3b, c and Supplementary Fig. 5a, b).

Comment 6. In line 265, Os1900 should be Os1400.

Response: Thank you for this comment, we have corrected it. Now it is in line 270.

Comment 7. In line 243, Fig. 3a should be Fig. 3a-b

Response: According to your comment, we have corrected it, now it is Fig.3b-c, since the previous Fig. 3 has been further modified.

Reviewer #3:

Summary:

The authors conducted transcriptomic analyses on rice plants grown under various different fertilizer treatments to identify key genes respond to fertilizer, independently of changing environmental conditions. They identified that Os1900, a MAX1-like gene involved in the strigolactone synthesis pathway, strongly responds to nutrient deficiency. By generating mutants in this gene and the closely related Os5100, they showed that both these genes are required to control rice tiller number. Tissue-specific expression of Os1900 in shoots alters SL-intermediates distribution compared to roots, indicating a possible way of controlling tissue-specific responses. Finally, the authors generate a series of mutants in the Os1900 promoter, producing rice plants with potential agronomical properties.

Response: Thanks for this concise summary of our work. We appreciate it very much. It is of note that the results of promoter mutant analyses also provide us with the related strong evidence, that is, *Os1900* transcriptional regulation is a key to control tiller numbers in rice in addition to possible creation of novel useful alleles in rice breeding.

Comments to authors:

Comment 1: *I find that the introduction section is lacking key references describing previous work that studies strigolactone involvement in plant development in response to nutrient deficiency. The authors discuss the role of SLs in indirect nutrient uptake via with arbuscular mycorrhizae symbiosis, but there is ample work showing how strigolactone-related genes are regulated in response to nutrient deficiency and more importantly, how they affect branching (eg., (Sun et al., 2014; Marzec et al., 2013; de Jong et al., 2014)).*

Response: Thanks for pointing out our carelessness. References of Booker et al. (2005), Marzec et al. (2013), and Sun et al. (2014) have been cited in introduction. Reference of de Jong et al. (2014) has been cited in the result section. Now their Ref. No. are 7, 8, 9, and 28.

Moreover, it is stated in lines 59-63 that “Fertilization results in biological changes in many plant growth parameters (...). However, the mechanisms by which fertilization controls these traits remain largely unknown”, while it is obviously clear that there is ample literature on this matter, with whole agronomy research fields dedicated to this matter. Please rephrase this sentence.

Response: Thanks for your comment. We have removed it and revised the sentence in our revised manuscript, lines 54-59.

Comment 2: Lines 120-122 and Figure 1c. A single Fold Change value is being reported. According to the legend, it seems like averaged fold change of Fertilized vs non-fertilized at each time point is being represented, but it is not sufficiently clear. Also, there are one/two fertilization samples, are both treatments also being averaged? Please clarify more in detail in the text/methods section.

Response: We appreciate your valuable comment. We have used the average values of log₂-transformed fold change values for all tested pairs of samples (none vs. one, none vs. two, and one vs. two) at each sampling point for each gene. We have clarified this in the captions, lines 138-142.

Comment 3: Lines 146-149. Although it might not be the scope of this paper, probably there is additional interesting information to be extracted from the transcriptomic analysis. Are there any other interesting fertilization-responding genes? At least, the response of other MAX1-like genes should be included in the text, even if they do not respond in the collected tissue (this would more strongly highlight the importance of Os1900 response in this tissue).

Response: Yes, as you pointed out, probably due to the tissue-specific expression, only *Os1900* was detected in our fertilizer response experiments. And the other *MAX1*-like genes and SL- related genes were not detected or had no obvious change, thus they were not found in the 107 fertilizer response genes. We have corrected this point and added a related description, line 148-150.

Comment 4: Lines 160-162. The authors state that they identified the Os1900 response “under normal rice cultivation conditions” but they measured its response comparing fertilized to non-fertilized conditions. Could the authors provide a rough estimate of P/N/K concentration in the soil from which these plants were grown? Probably P/N/K concentration was low in non-fertilized samples explaining why Os1900 was induced in field conditions.

Response: Thanks for this valuable comment. In the paddy field cultivation conditions for field transcriptome analysis, we used ~3kg N/10a, ~4kg P/10a, ~5kg K/10a for both the first and the second fertilization, according to a conventional cultivation protocol in Japan. We have added this information in the related captions, lines 532-534. Since we use the same paddy fields every year, we did not consider the residual fertilizer from previous cultivations on our non-fertilizer areas in this experiment. We agree with the reviewer’s idea, in which P/N/K concentration was low in the non-fertilized area.

Comment 5: Lines 163-171 and Figure 2d. The authors describe that only the Os1900 Os5100 double mutant shows a tillering phenotype. It is surprising that the single Os1900 mutant has a wild-type tillering phenotype. If Os1900 is induced under nutrient deficiency conditions, one would expect that the single Os1900 mutant would show a significant phenotype in this condition. The authors show in Supplementary Figure 3a that this is not the case.

These results suggest that maybe there is a compensatory mechanism by which Os5100 is induced in the absence of Os1900. I suggest that the authors measure Os5100 mRNA levels (and maybe other MAX1-like genes) in single Os1900 mutants and vice-versa, under normal and nutrient deficiency conditions to clarify this question.

Response: Thanks for this valuable comment. It is a very important point. Accordingly, we first grew *os1900* single, *os5100* single mutants under minor- and trace-fertilization conditions, and found that *os1900* exhibited a slight tillering advantage under trace-fertilizer conditions (Fig. 4c, d). We further examined the expression of *MAX1*-like genes in *os1900* single mutant and *os5100* single mutant under trace- and no-fertilizer conditions. Interestingly, in the shoot samples, we found that *os1900* mutations did not have any effects on the expression of *Os5100* (Supplementary Fig. 3c). Similarly, although the loss of *Os900* and *Os1400* led to increased tillers (Cardoso et al.2014), it

has been recently reported that in *os900* single mutant, the expression of other *MAX1*-like genes (*Os1900*, *Os5100*, *Os1400*) was not changed (Ito et al.2022). How *MAX1*-like genes are related to each other to produce SLs and to control tiller numbers is still a very complex problem.

I would also suggest to include a quantification of the tillering phenotype shown in supplemental figure 3. While the image might be representative, only a single picture is shown.

Response: The number of tillers under the condition of no fertilizer is added in Supplementary Fig. 3a, b.

Comment 6: Lines 242-243. *The authors state that “Significant CL accumulation was observed in *os1900*&*os5100* double mutants in both shoots and roots” while the results show high variability in the quantification of CL in roots and CLA in shoots, with p-values showing low statistical significance. If the authors want to show statistically significant changes, I would suggest increasing n. Alternatively, I would suggest to rephrase the text.*

Response: Accordingly, we have repeated SL measurements with more biological replicates (n=5), and revised the text in our manuscript, lines 237-253.

Comment 7: Lines 289-290 *“Tiller elongation is inhibited at axial bud regions by SLs; however, it remains unknown whether SLs can migrate to action sites.” I find it quite surprising that the authors make this statement. In fact, intermediate products of SLs were described as mobile signals before SL structure, synthesis and signaling were known (Booker et al., 2005). More specifically, MAX1 was identified as a key gene able to process the SL intermediate signal in axillary buds to produce a branch inhibiting SL.*

Response: We have removed it and corrected sentences based on your suggestion, lines 343-344.

Comment 8: Lines 361-363. *In the experiments with *Os1900* promoter mutant plants,*

the authors state that “Rates of decrease were low in M3, M6, and M9”. These lower rates of decrease could be also partially explained by lower “starting” expression in 30 ppm 0h conditions (although other mutants still show a “WT” decrease in Os1900 levels). In the text it is stated that “M3, M6 and M9 contain critical cis-regulatory elements for fertilizer responses” while it seems to be the contrary, they seem to contain regulatory elements needed for Os1900 induction in response to nutrient deficiency. Moreover, they do not seem to be “critical” as there is still some degree of response. In fact, it seems that mutants with lower Os1900 induction at 30 ppm 0h show the strongest phenotype in non-fertilized conditions, indicating that Os1900 levels correlate directly with the tiller number. Could the authors represent Os1900 expression levels with tiller number? Is there a direct relationship? As the authors suggest, this might not be the case (as M4, M6, M7, M8, and M9 mutants produce more tillers than WT), but I think it would be interesting to show the data to back up this conclusion. Also, in lines 368 to 388, conserved CREs in other SL genes are studied, while these genes show clearly different expression patterns (shoot vs root) and different responses to nutrients. To me, the conservation of these motifs and their location in the promoter does not add any meaningful information. Perhaps only the closest MAX1-like genes should be included in this analysis.

Response: We are grateful for your comments.

1) About *M3*, *M6* and *M9*:

M3, *M6*, and *M9* mutants with more gentle slopes than WT, exhibited less responsive to fertilizer at 30ppm. This indicates some critical cis-regulatory elements for the fertilizer responses in these deletion regions, see Supplementary Fig. 11 and Supplementary Table 8.

2) About *Os1900* expression and tillers:

Among all the mutants, *M4* had the lowest expression of *Os1900* and showed more tillers and higher grain yields under minor fertilization conditions. However, the effect of cis-regulation variation on traits is not a simple linear relationship with tiller numbers and elongations. (as also reported by Rodríguez-Leal D, 2017).

3) About CREs:

According to your comments, we have removed that of *D10*, *D17*, *D27* and reserved *MAX1*-like genes (Fig. 5d, e). In addition, we focused on the P1BS motif, the CREs for the Pi starvation response, which can interact with two TFs of *PHR1* and *PHL1*.

In the *Os1900* promoter region, four P1BSs were found. In addition, the deletion region of *M4* mutant contained one P1BS site; the *Os1900* expression decreased in *M4* mutant under the four fertilization conditions (50ppm 0h and 1h; 30ppm 0h and 1h). Under minor fertilizer conditions, *M4* presented increased tillers and grain yields (Fig. 6, Supplementary Fig.12).

Comment 9: The authors suggest that the increased tiller number in some Os1900 promoter mutants “may result in higher panicle numbers and yields in rice cultivation”. I think this is a key missing result. Could the authors quantify panicle number/yield of these mutants to show if this is true?. Also, although Os1900 seems to be mainly expressed in shoots, it would be important to show if those mutants have any effect on AM symbiosis. Is the change in tiller number just a consequence of different strigolactone processing in shoots or is indirect nutrient uptake involved?

Response: Thank you for your comments.

- 1) We investigated grain yields of each promoter mutant under laboratory cultivation conditions and found that the grain yield in *M4* mutant increased significantly (Supplementary Fig. 12). Since CRISPR/Cas9 mutants are not allowed to be cultivated in the fields in Japan without formal processes guided by Japanese governments, we would be only able to conduct the detailed field experiments on this part in the future.
- 2) PHRs are required for mycorrhizal symbiosis and regulate symbiosis-related genes via P1BS motif. *Os1900* is highly expressed in shoots and roots. We found four P1BS in the *Os1900* promoter region. The *M4* mutant contains one P1BS site, thus, likely lose the binding of PHR, thereby affecting AM symbiosis, but we did not check any experiments, thus no description in the revised manuscript at this moment, Lines 473-477.

Minor comments:

Comment 1: I suggest including the treatment diagram shown Supplementary figure 1a into the main figure 1. It would make it much easier for the reader to comprehend the sampling procedure. Also, explaining it in more detail in the manuscript would make it easier to understand.

Response: Accordingly, we have corrected and moved it to a main Figure 1a and explained it in the methods with more details, lines 131-134.

Comment 2: Please change the colors of Supplementary Figure 1d. In its current state it is almost impossible to follow the legend, as there are duplicated colors/shapes. One idea could be using empty/filled shapes for morning and afternoon samples, if indeed they PCA splits the samples between these factors.

Response: Thank you for your comments. We have corrected it as show now in Supplementary Fig. 1c.

Comment 3: Please change order of figure 3. Panel h should be panel a, and cited in lines 220-226. Also, figures 3 and 4 are not cited in order in the text, making it harder to follow.

Response: We have corrected the Fig. 3 by taking the previous h to a, and cited it, line 216. The previous Figs. 3, 4 had been merged into the current Fig.3, which was cited in text.

Comment 4: Lines 205-206 and Supp figure 4. Please explain more in detail the growth conditions of this experiment. In the text it is stated that the authors “compared the effects of fertilization” but it seems that plants were grown only under one condition. Were they fertilized? Were they under nutrient deficiency?

Response: Thank you for your comments. They were grown under normal fertilization conditions. We have added this information in the caption of Supplementary Fig.4.

Comment 5: In Figure 4b, I assume the legend refers to 1h of fertilization, but it is not stated.

Response: To better explain the relationship between expression of *MAX1* genes and SLs result, we removed the previous Fig. 4b, and detected the expression of *Os900*, *Os1400*, *Os1900* and *Os5100* using the same samples with SLs analysis. They are shown in Fig. 3d, e.

Comment 6: Throughout the whole paper tiller number phenotyping is conducted with what seems to be a very low population. In the figures where n is reported it varies from 4 to 6, and in some cases, it seems that n equals 3. Are these individual plants? Are these averaged experiments of larger populations? Please clarify in the figure legends and/or methods section.

Response: Thank you for your comments. For real-time PCR, n=3 or 4. They are 3 or 4 biological replates, including more than 9 individuals. Leaves and shoots from three plants or seedling bases from five individuals were pooled as a single sample for real-time PCR, with three or four biological replicates, lines 568-570.

For the observation of tiller numbers, we have increased the biological replicate numbers to 5-8 (Fig.4c, d; Fig.6).

Reviewer #2 (Remarks to the Author):

In this revised manuscript, the authors have performed useful additional experiments and addressed many of the specific experimental criticisms raised by reviewers. The reviewer appreciates these valuable efforts. The quality of experiment data and manuscript have been substantially improved, but the regulatory roles and molecular mechanisms of Os1900, Os5100 and other OsMAX1 in SL biosynthesis, tiller development and nutrient response still need improvement.

The following points need to be further addressed:

1. The authors draw the conclusion that fertilization controls tiller number in rice through transcriptional regulation of Os1900. The evidence to support this core conclusion is not strong enough. First, among the known SL biosynthesis genes, only Os1900 was in the list of 107 genes, which were identified as stable fertilizer-responsive genes in rice leaves under natural paddy field conditions through transcriptome analysis. This is probably due to the tissue specificity of gene expression. In our experimental system, the expression levels of D10, D17, D27, Os900 and Os1400 in roots were all greatly induced in roots under Pi deficiency. In the shoot base, the induction by Pi deficiency was greatly weakened (for D10 and D17) or abolished (for D27, Os900, Os1400), indicating that these genes are important for the induction of SL biosynthesis under Pi deficiency in roots. Consistently, the authors found that Os900 and Os1900 contained many overlapping CREs, including P1BS. How do Os900 and Os1900 function differently in response to fertilization.

Second, the Os900 Os1400 double mutant formed more tillers than the wild type, although the elevation was not as high as that of the Os1900 Os5100 double mutant. This observation is under normal growth conditions. To observe their roles in tiller response to fertilization, the authors are suggested to compare the tiller number of WT, Os900 Os1400 double mutant and Os1900 Os5100 double mutant under different fertilization conditions.

2. Another important point is about the novelty of this research. As mentioned in Lines 218-220, the CL feeding assay using recombinant proteins in yeast suggested that Os1900 and Os5100 can catalyze the conversion of CL to CLA (Ref18). The observation of CA and CLA levels in the os1900 os5100 double mutant is consistent with the roles of Os1900 and Os5100 in yeast, but the conceptual breakthrough is not strong. Furthermore, the 4DO levels in roots of os1900 os5100 double mutant were higher than WT, but the authors did not give a reasonable explanation. Although CLA production is defective in os1900 os5100 double mutant, the direct effects of CLA in tiller development still need further study, because it is not clear whether CLA treatment can rescue the high tillering phenotype of os1900 os5100 double mutant.

3. Notably, the authors performed RNA-seq analysis and try to investigate the mechanism underlying tillering development of os1900 os5100. In addition to the transcriptional changes in D53 and OsTB1, various transcriptional regulations related to the biosynthesis and signaling of other phytohormones were observed, but it is difficult to evaluate which pathways are important for tiller regulation in os1900 os5100. For instance, the authors mentioned that "indicating that CKs is the hormone for promoting tiller elongation" in lines 329-330. Based on the expression levels of CK biosynthesis genes and CK inactivation genes, this inference is inaccurate, unless the CK levels are measured. Similarly, the contents of "FT-like genes i.e. Hd3a in rice 39 and PtFT1 in Populus tomentosa 40, 41 could promote the lateral branching. Thus, we estimated that the rice FT-like gene of FTL6 was involved in breaking axillary bud dormancy in rice, because it expressed in a higher level in leaves in os1900&os5100 than that in WT (Supplementary Fig. 9b)." in lines 355-357 are also less rigorous.

4. What is the reason for the elevated grain yield in the M4 mutant under minor fertilization conditions? The M3, M4 and M2 mutants formed more tillers under minor fertilization, but M3 and M2 showed comparable grain yields to the WT. The authors are suggested to give some explanation about this important result with potential application in agriculture.

5. The stable fertilizer-responsive genes (107 genes) in rice leaves under natural paddy field conditions through transcriptome analysis are suggested to clearly list in a Supplementary table.

Reviewer #3 (Remarks to the Author):

The authors have improved the manuscript by increasing the detail of some sections of the paper, performing additional experiments and increasing sample size and statistical analysis according to the editor/reviewer's suggestions. I appreciate their additional work and I think this revised version of the article is stronger and more grounded in the data.

I have some comments to some of the author's responses and the new manuscript version.

Response #11:

1) About M3, M6 and M9:

M3, M6, and M9 mutants with more gentle slopes than WT, exhibited less responsive to fertilizer at 30ppm. This indicates some critical cis-regulatory elements for the fertilizer responses in these deletion regions, see Supplementary Fig. 11 and Supplementary Table 8.

Lines 467-470 still state that there are "some critical cis-regulatory elements in these deletion regions for the fertilizer responses". This is not conceptually correct. These mutants show no fertilizer response because their Os1900 expression levels are already low in non-fertilizer conditions. If their levels were still high after fertilization, that would mean that Os1900 does not respond to fertilizer application. This could be rephrased as "they seem to contain regulatory elements needed for Os1900 induction in response to nutrient deficiency.", although I think that would make it more confusing.

3) About CREs:

According to your comments, we have removed that of D10, D17, D27 and reserved MAX1-like genes (Fig. 5d, e). In addition, we focused on the P1BS motif, the CREs for the Pi starvation response, which can interact with two TFs of PHR1 and PHL1. In the Os1900 promoter region, four P1BSs were found. In addition, the deletion region of M4 mutant contained one P1BS site; the Os1900 expression decreased in M4 mutant under the four fertilization conditions (50ppm 0h and 1h; 30ppm 0h and 1h). Under minor fertilizer conditions, M4 presented increased tillers and grain yields (Fig. 6, Supplementary Fig.12).

I think the P1Bs results are inconclusive and do not provide any link between P1HB sites and Os1900 regulation, and thus do not add relevant information to the paper.

Additional comments:

From the PCA presented in SFig 1C, the main factor affecting variability is clearly sampling time. There is slight clustering based on transplantation time, but with a quite high overlap, and fertilization regime does not seem to cluster samples at all. This does not necessarily mean that fertilization does not have an effect, but that its effect might be specific to some genes, not to the whole transcriptome.

Line 150-151 "However, only Os1900 were responsive to normal fertilizer application critically in the leaf samples under paddy field cultivation conditions."

I would clearly state that Os1900 expression is repressed by fertilization. Although it is clearly shown later in the paper, this would make it easier to the reader to follow the logic of the work.

Line 280-282 "which indicates some control by Os1900 or Os5100 in producing MeCLA through CLA methylation".

I think the way this sentence is written might be confusing, as it could be interpreted as if the authors are implying that Os1900 or Os5100 are involved in CLA methylation. If there is no CLA in these mutants, the lack of MeCLA can be directly explained by the absence of CLA, its precursor.

Line 489-491. "Although the simple defect mutation of Os1900 resulted in no increase in tiller numbers in this study, we further investigated tiller number dynamics in these promoter mutants under minor and trace fertilization conditions".

Please correct this sentence, as Os1900 does show increased tiller in some conditions (Fig. 4d).

Lines 497-501. (...)

The yield increase in M4 lines cannot be directly related to the tiller phenotype, as other lines do have similar tiller increase (e.g., M3) and no yield increase, while at the same time, other lines do not have a tiller number change and lower yield, such as M7. I understand the technical limitations of not being able to produce field data with CRISPR-edited lines that the authors have mentioned, but I would interpret these yield results with much more caution than the authors have done in the text.

Lines 513-515. "Although further detailed CRE identification is still essential, our results clearly indicate that fertilization controls tiller numbers via transcriptional regulation of the Os1900 gene".

Given the evidence shown in this work, there is not a clear direct correlation between Os1900 levels/response and tiller number. Lines M3, M4 and M6 show increased tiller number, while their Os1900 levels/response are quite different. To me, this is not a clear indication.

Lines 521-523. "and demonstrated that Os1900 transcriptional regulation induced by fertilizers is a key in controlling the tiller number and yield for rice."

Again, I don't think the authors have demonstrated that tiller number can be directly correlated with yield. The only example they show is M4, but other mutants with increased tiller number do not show increased yield (e.g, M3 or M6).

The point-by-point responses to the reviewers' comments

Manuscript ID: NCOMMS-22-05588A

Title: Fertilization controls tiller numbers via transcriptional regulation of a *MAX1*-like gene in rice cultivation

Dear our editor & all the reviewers,

Thanks for taking care of our revised manuscript. This time, we have added temporal growth data of tiller number for *os900* & *os1400* under minor- and trace- fertilization conditions (Supplementary Figure 4d) according to the reviewer's comments and fertility data of our promoter deletion mutants (Supplementary Figure 12b) in our newly revised manuscript to better explain the role of *Os1900* in controlling tiller numbers and the reason why only *M4* mutant showed superiority in yield, respectively.

Reviewer #2 (Remarks to the Author):

*In this revised manuscript, the authors have performed useful additional experiments and addressed many of the specific experimental criticisms raised by reviewers. The reviewer appreciates these valuable efforts. The quality of experiment data and manuscript have been substantially improved, but the regulatory roles and molecular mechanisms of *Os1900*, *Os5100* and other *OsMAX1* in SL biosynthesis, tiller development and nutrient response still need improvement.*

Response: Thanks for this appreciation for our efforts to improve the manuscript and valuable further comments.

The following points need to be further addressed:

*1. The authors draw the conclusion that fertilization controls tiller number in rice through transcriptional regulation of *Os1900*. The evidence to support this core conclusion is not strong enough. First, among the known SL biosynthesis genes, only *Os1900* was in the list of 107 genes, which were identified as stable fertilizer-responsive genes in rice leaves under natural paddy field conditions through transcriptome analysis. This is probably due to the tissue specificity of gene expression. In our experimental system, the expression levels of *D10*, *D17*, *D27*, *Os900* and *Os1400* in roots were all greatly induced in roots under Pi deficiency. In the shoot base, the induction by Pi deficiency was greatly weakened (for *D10* and *D17*) or abolished (for *D27*,*

Os900, Os1400), indicating that these genes are important for the induction of SL biosynthesis under Pi deficiency in roots. Consistently, the authors found that *Os900* and *Os1900* contained many overlapping CREs, including *PIBS*. How do *Os900* and *Os1900* function differently in response to fertilization.

Response: Thank you very much for your comments. We totally agree that the reason why only *Os1900* was identified in our study would be probably due to the tissue specificity of gene expression, and the gene induction by Pi deficiency was stronger in root than in shoot, suggesting that roles of SLs biosynthesis genes including *Os1900* may be important under Pi deficiency in roots. It is also of note that in this manuscript, we do not deny the importance of *Os1900* expression in roots and/or other regions under various environmental conditions at all.

Importantly, using the double mutants, we have demonstrated that *Os1900* & *Os5100* encode major P450 enzymes to produce CLA from CL and contribute to the tiller number control in rice more than *Os900* & *Os1400*. In addition, some promoter deletion mutants in *Os1900*, which change only the expression pattern (place, timing, and conditions) of *Os1900* transcripts, caused the change of tiller numbers. Thus, these data could be enough to conclude the importance of transcriptional regulation of *Os1900*. We should tell that further *in situ* analysis of detailed gene expression in those deletion mutants is going in our laboratory. We think that we need such results to explain why *Os1900* function differently from *Os900* although we guess the *Os1900* expression pattern in shoots by fertilizer may mean something to confer the difference. Indeed, as a related knowledge we know, Mashiguchi et al, 2022 showed that in *Arabidopsis thaliana*, the grafted plant with *max4* as a scion and *max1* as a rootstock didn't show more tillers, which meant that CL from roots moved to shoots then converted to SLs to inhibit branching in shoots. This suggests that the biosynthesis of SLs in shoots is also important to control shoot branching.

Furthermore, *Os900* and *Os1400* are proposed to be inserted to the rice *japonica* genome relatively recently (Qi et al, 2021). Under Pi starvation conditions, *Os900* is still induced in roots, but weakened or not expressed in shoots, maybe due to a compensation mechanism. *Os1900* is induced in both roots and shoots, which may imply its irreplaceable role in rice.

Second, the Os900 Os1400 double mutant formed more tillers than the wild type, although the elevation was not as high as that of the Os1900 Os5100 double mutant. This observation is under normal growth conditions. To observe their roles in tiller response to fertilization, the authors are suggested to compare the tiller number of WT, Os900 Os1400 double mutant and Os1900 Os5100 double mutant under different fertilization conditions.

Response: Thanks for your comment. Accordingly, we have added the growth of tiller numbers in *os900&os1400* under minor and trace fertilization conditions (Supplementary Fig. 4d). The results show almost the same tiller numbers with the wild type. These results are described in the main text in line 206-207.

2. Another important point is about the novelty of this research. As mentioned in Lines 218-220, the CL feeding assay using recombinant proteins in yeast suggested that Os1900 and Os5100 can catalyze the conversion of CL to CLA (Ref18). The observation of CA and CLA levels in the

os1900 os5100 double mutant is consistent with the roles of Os1900 and Os5100 in yeast, but the conceptual breakthrough is not strong.

Response: We know that the previous data of SLs biosynthesis did not deny the importance of *Os1900* and *Os5100*. Ref18 shows that both *Os1900* and *Os5100* can catalyze the conversion of CL to CLA. However, these experiments were conducted in yeast (nearly *in vitro*).

In our study, the double mutations in both *Os1900* and *Os5100* leads to the accumulation of a large amount of CL *in vivo*, fully affirm the important role of *Os1900* and *Os5100* cooperated to convert CL into CLA and control tillers genetically. It is of note that those previous data also suggested us that *Os900* and *Os1400* are more important than *Os1900* and *Os5100* for controlling tillering. In addition to this, with data in the QTL analysis (Cardoso et al, 2014), people in this field believe that *Os900* and *Os1400* genes are major players in tiller number control. Now, we believe that our study will change this situation to the right direction. In addition, in this work, we have succeeded to demonstrate that transcriptional regulation of *Os1900* is an important key to control tiller numbers via fertilizers in rice and create new useful alleles of promoter deletion mutants for new rice cultivars in the future sustainable development goals era. We believe that the novelty of our study is quite enough. Of course, we agree that, unfortunately, due to the particularity of SL biosynthesis pathway in rice and experimental period, we cannot carry out stable isotope labeling experiments by feeding plants with ¹³C-CL (CL = carlactone) which was mentioned before. Feeding plants with ¹³C-CL, one could detect ¹³C-labeled CLA (CLA = carlactonoic acid) in wild type (WT), but not in *os1900* & *os5100* double mutants. However, the ¹³C-labelled target compound would be largely diluted since plants can produce endogenous CL by themselves. To address this issue, it is essential to compare a *d10* rice mutant (make rice plants impossible to produce endogenous CL) and *d10* + *os1900* + *os5100* triple mutants. However, we do not have *d10* mutant in the Koshihikari genetic background, *d10* + *os1900* + *os5100* triple mutants, and ¹³C-labelled CL in our labs. These synthesis and preparation would take more than one year. We are sorry to give up this plan finally in this revision process.

Furthermore, the 4DO levels in roots of os1900 os5100 double mutant were higher than WT, but the authors did not give a reasonable explanation. Although CLA production is defective in os1900 os5100 double mutant, the direct effects of CLA in tiller development still need further study, because it is not clear whether CLA treatment can rescue the high tillering phenotype of os1900 os5100 double mutant.

Response: Thank you very much for your comments. We have added some explanation of the higher 4DO level in root of *os1900&os5100*, line 272-274. We think that the over-accumulated CL in roots of *os1900&os5100*, would lead to high production of 4DO by Os900.

As for the CLA application experiment, because it needs a large amount of CLA, we regret that we can't carry it out. However, it has been reported in *Arabidopsis thaliana* that CLA can inhibit shoot branching in a CLA-deficient mutant (Abe et al. 2014)

3. Notably, the authors performed RNA-seq analysis and try to investigate the mechanism underlying tillering development of os1900 os5100. In addition to the transcriptional changes in D53 and OsTB1, various transcriptional regulations related to the biosynthesis and signaling of

other phytohormones were observed, but it is difficult to evaluate which pathways are important for tiller regulation in os1900 os5100. For instance, the authors mentioned that “indicating that CKs is the hormone for promoting tiller elongation” in lines 329-330. Based on the expression levels of CK biosynthesis genes and CK inactivation genes, this inference is inaccurate, unless the CK levels are measured. Similarly, the contents of “FT-like genes i.e. Hd3a in rice 39 and PtFT1 in Populus tomentosa 40, 41 could promote the lateral branching. Thus, we estimated that the rice FT-like gene of FTL6 was involved in breaking axillary bud dormancy in rice, because it expressed in a higher level in leaves in os1900&os5100 than that in WT (Supplementary Fig. 9b).” in lines 355-357 are also less rigorous.

Response: Thanks for these comments. We agree that it is difficult to evaluate which pathways are more important for tiller regulation at this moment. The data of RNA-SEQ was a kind of surprise to us, too. We did not expect such a central role for SLs- biosynthesis to affect various biological pathways. In this manuscript, we would like to report this central role of SLs- production at least. Then, further analysis on the genes which were affected in the *os1900* & *os5100* double mutants will be planned in the future and become an open question in this field. The feedback effects related to SLs-biosynthesis deficiency will be also our targets. Thank you for pointing out the following mistakes, too.

1. About “*indicating that CKs is the hormone for promoting tiller elongation*”.

We used the wrong word of “*indicating*”, which is confusing. In fact, “*CKs is the hormone for promoting tiller elongation*” is not our conclusion. It is the conclusion of Ref34. Now we have corrected it in line 334.

It should be that previous research has reported that CK is a hormone that promotes tillering. In our *os1900&os5100* double mutant with more tillers, the expression of CKs biosynthesis genes increases, and the expression of CKs inactivation- related genes decreases. This is consistent with previous studies such as Ref34.

2. About “*FT-like genes i.e. Hd3a in rice (39) and PtFT1 in Populus tomentosa (40, 41) could promote the lateral branching. Thus, we estimated that the rice FT-like gene of FTL6 was involved in breaking axillary bud dormancy in rice, because it expressed in a higher level in leaves in os1900&os5100 than that in WT (Supplementary Fig. 9b)*”, we have corrected this part, check line 359-366.

4. What is the reason for the elevated grain yield in the M4 mutant under minor fertilization conditions? The M3, M4 and M2 mutants formed more tillers under minor fertilization, but M3 and M2 showed comparable grain yields to the WT. The authors are suggested to give some explanation about this important result with potential application in agriculture.

Response: Thanks for pointing it out. We have added the data on seed fertility (Supplementary figure 12b). Due to the high seed sterility of *M3* and *M6*, the yield decreased. We still do not know whether the high sterility is related to the deletions or whether other extra unexpected mutations occurred in the tested lines. We are now planning some further genetic analysis using the mutants whether we can segregate the phenotypes or not.

5. The stable fertilizer-responsive genes (107 genes) in rice leaves under natural paddy field conditions through transcriptome analysis are suggested to clearly list in a Supplementary table.

Response: The 107 genes have been listed in the Supplementary table3. Please check it.

Reviewer #3 (Remarks to the Author):

The authors have improved the manuscript by increasing the detail of some sections of the paper, performing additional experiments and increasing sample size and statistical analysis according to the editor/reviewer's suggestions. I appreciate their additional work and I think this revised version of the article is stronger and more grounded in the data.

Response: Thank you very much for your appreciation of our work.

I have some comments to some of the author's responses and the new manuscript version.

Response #11:

1) About M3, M6 and M9:

M3, M6, and M9 mutants with more gentle slopes than WT, exhibited less responsive to fertilizer at 30ppm. This indicates some critical cis-regulatory elements for the fertilizer responses in these deletion regions, see Supplementary Fig. 11 and Supplementary Table 8.

Lines 467-470 still state that there are "some critical cis-regulatory elements in these deletion regions for the fertilizer responses". This is not conceptually correct. These mutants show no fertilizer response because their Os1900 expression levels are already low in non-fertilizer conditions. If their levels were still high after fertilization, that would mean that Os1900 does not respond to fertilizer application. This could be rephrased as "they seem to contain regulatory elements needed for Os1900 induction in response to nutrient deficiency.", although I think that would make it more confusing.

Response: Thank you for pointing it out. I have corrected as you said, line 473-474.

3) About CREs:

According to your comments, we have removed that of D10, D17, D27 and reserved MAX1-like genes (Fig. 5d, e). In addition, we focused on the P1BS motif, the CREs for the Pi starvation response, which can interact with two TFs of PHR1 and PHL1. In the Os1900 promoter region, four P1BSs were found. In addition, the deletion region of M4 mutant contained one P1BS site; the Os1900 expression decreased in M4 mutant under the four fertilization conditions (50ppm 0h and 1h; 30ppm 0h and 1h). Under minor fertilizer conditions, M4 presented increased tillers and grain yields (Fig. 6, Supplementary Fig.12).

I think the PIBs results are inconclusive and do not provide any link between PIHB sites and Os1900 regulation, and thus do not add relevant information to the paper.

Response: We agree that the P1BS results are inconclusive and do not provide any strong link between P1BS sites and Os1900 regulation. However, P1BS often exist in the promoter of Pi starvation-responsive structural genes. It can be said that it has one of the highest credibility among all predicted CREs related to fertilization. Thus, we have changed the description on the

relationship between our results of deletion mutants and P1BS sites in *Os1900* promoter carefully. Check it in line 489-495.

Additional comments:

From the PCA presented in SFig 1C, the main factor affecting variability is clearly sampling time. There is slight clustering based on transplantation time, but with a quite high overlap, and fertilization regime does not seem to cluster samples at all. This does not necessarily mean that fertilization does not have an effect, but that its effect might be specific to some genes, not to the whole transcriptome.

Response: Yes, we agree with you. We have changed related expression according to this comment. Please check it in line 121-124.

*Line 150-151 “However, only *Os1900* were responsive to normal fertilizer application critically in the leaf samples under paddy field cultivation conditions.”*

*I would clearly state that *Os1900* expression is repressed by fertilization. Although it is clearly shown later in the paper, this would make it easier to the reader to follow the logic of the work.*

Response: We have corrected as you wrote. See Line153-155.

*Line 280-282 “which indicates some control by *Os1900* or *Os5100* in producing MeCLA through CLA methylation”.*

*I think the way this sentence is written might be confusing, as it could be interpreted as if the authors are implying that *Os1900* or *Os5100* are involved in CLA methylation. If there is no CLA in these mutants, the lack of MeCLA can be directly explained by the absence of CLA, its precursor.*

Response: We have corrected it as you wrote. See line 286-289.

*Line 489-491. “Although the simple defect mutation of *Os1900* resulted in no increase in tiller numbers in this study, we further investigated tiller number dynamics in these promoter mutants under minor and trace fertilization conditions”.*

*Please correct this sentence, as *Os1900* does show increased tiller in some conditions (Fig. 4d).*

Response: Thank you very much, we have corrected it. See line 498.

Lines 497-501. (...)

The yield increase in M4 lines cannot be directly related to the tiller phenotype, as other lines do have similar tiller increase (e.g., M3) and no yield increase, while at the same time, other lines do

not have a tiller number change and lower yield, such as M7. I understand the technical limitations of not being able to produce field data with CRISPR-edited lines that the authors have mentioned, but I would interpret these yield results with much more caution than the authors have done in the text.

Response: Thank you for your comments. We agree with you. We have added the fertility data (Supplementary figure 12b). Due to some unknown reasons, the fertility decreased in *M3*, *M6*, and *M7*. *M4* is the sole mutant with more tillers and higher yield so far. Accordingly, we have changed the interpretation of our yield results. See line 505-509.

Lines 513-515. "Although further detailed CRE identification is still essential, our results clearly indicate that fertilization controls tiller numbers via transcriptional regulation of the Os1900 gene".

Given the evidence shown in this work, there is not a clear direct correlation between Os1900 levels/response and tiller number. Lines M3, M4 and M6 show increased tiller number, while their Os1900 levels/response are quite different. To me, this is not a clear indication.

Response: Thanks for your comments. We have corrected it in line 528-529. The relationship between the expression of target genes in promoter mutants and phenotype is not simply linear. Similar results have also been reported in tomato (Rodríguez-Leal et al, 2017). We believe that we need more detailed expression analysis including *in situ* hybridization approach under various cultivations. At this moment, we think that more important thing is the fact we have created a few possibly useful alleles among various promoter deletion mutants. Especially, *M4* mutant increased both tiller numbers and the yield. Not perfectly but consistently, the expression of *Os1900* in this *M4* mutant is lower than WT under different fertilization conditions.

Lines 521-523. "and demonstrated that Os1900 transcriptional regulation induced by fertilizers is a key in controlling the tiller number and yield for rice."

Again, I don't think the authors have demonstrated that tiller number can be directly correlated with yield. The only example they show is M4, but other mutants with increased tiller number do not show increased yield (e.g, M3 or M6).

Response: We have corrected it. See line 505-509 and 528-529.

Now we have added the data of seed fertility (Supplementary figure 12b), which can better explain the results to reduce the confusion. There are many factors affecting yields, and tiller number is one of them. Then, indeed, *M4* line increased the yield in our laboratory. Other mutants with tiller number increase may have some problems, especially seed fertility, where we are still not sure about the genuine genetic reasons. Although we didn't mean that the yield would increase with more tillers, but the phenotypes of *M3* and *M6* with more tillers may mean something in the future in addition to the promising *M4* case.

Reviewer #2 (Remarks to the Author):

The revised manuscript is further improved. The authors have performed useful experiments and added reasonable explanation. I appreciate their efforts. I am satisfied with the authors' response.

Reviewer #3 (Remarks to the Author):

The authors have responded accordingly to my comments and modified the manuscript. I have no further questions.